# An improved organ explant culture method reveals stem cell lineage dynamics in the adult *Drosophila* intestine

Marco Marchetti, Chenge Zhang, Bruce A Edgar*

Department of Oncological Sciences, Huntsman Cancer Institute, University of Utah, Salt Lake City, United States

**Abstract** In recent years, live-imaging techniques have been developed for the adult midgut of *Drosophila melanogaster* that allow temporal characterization of key processes involved in stem cell and tissue homeostasis. However, these organ culture techniques have been limited to imaging sessions of ≤16 hours, an interval too short to track dynamic processes such as damage responses and regeneration, which can unfold over several days. Therefore, we developed an organ explant culture protocol capable of sustaining midguts ex vivo for up to 3 days. This was made possible by the formulation of a culture medium specifically designed for adult *Drosophila* tissues with an increased $Na^+/K^+$ ratio and trehalose concentration, and by placing midguts at an air-liquid inter-face for enhanced oxygenation. We show that midgut progenitor cells can respond to gut epithelial damage ex vivo, proliferating and differentiating to replace lost cells, but are quiescent in healthy intestines. Using ex vivo gene induction to promote stem cell proliferation using $Ras^{G12V}$ or *string* and *Cyclin E* overexpression, we demonstrate that progenitor cell lineages can be traced through multiple cell divisions using live imaging. We show that the same culture set-up is useful for imaging adult renal tubules and ovaries for up to 3 days and hearts for up to 10 days. By enabling both long-term imaging and real-time ex vivo gene manipulation, our simple culture protocol provides a powerful tool for studies of epithelial biology and cell lineage behavior.

*For correspondence:
bruce.edgar@hci.utah.edu

**Competing interest:** The authors declare that no competing interests exist.

## Editor's evaluation

Marchetti and colleagues present a promising, ex vivo culture method for the adult *Drosophila* midgut and other abdominal organs. Highlights include demonstrated organ viability for up to 72 hours and protocols for ex vivo injury and genetic manipulation. These advances enable the first real-time lineages tracking multiple stem cell divisions, which reveal intriguing spatial-temporal behavior patterns. The manuscript provides a thorough and thoughtful evaluation of these dynamic data and includes additional information about imaging parameters that will be invaluable to those seeking to replicate this method.

## Introduction

Endo- and ectodermal epithelia comprise essential interfaces between an organism and its environ-ment. As such, they form a first line of defense that is frequently subjected to diverse types of insult. This situation requires that epithelia be able to mount appropriate responses. This is possible in part due to the action of resident stem cells which, through their ability to self-renew and produce differen-tiated progeny, allow epithelia to regenerate both structurally and functionally. The adult *Drosophila melanogaster* midgut is a prime example of this as its population of intestinal stem cells (ISCs) are able to interpret signals from their surrounding environment such as cytokines released by neighboring

damaged enterocytes (EC) (*Ohlstein and Spradling, 2006*; *Micchelli and Perrimon, 2006*; *Jiang et al., 2009*; *Beebe et al., 2010*; *Patel et al., 2019*). When this interaction occurs, normally quiescent stem cells rapidly respond to the needs of the tissue, proliferating and stimulating the differentiation of their progeny to replace lost cells, thus repairing the damaged epithelium (*Jiang et al., 2009*; *Patel et al., 2019*).

The understanding of epithelial biology has been greatly advanced by protocols for the ex vivo culture and imaging of tissues and organs. For example, mammalian intestinal organoids have advanced the field by easily allowing the direct observation of stem cell behavior, without the need for intravital imaging (*Sato et al., 2009*). Several protocols have been developed for the live-imaging of *Drosophila* tissues and organs such as imaginal discs (*Robb, 1969*; *Zartman et al., 2013*; *Handke et al., 2014*; *Tsao et al., 2016*; *Strassburger et al., 2017*), larval brains (*Siller et al., 2005*; *Rabinovich et al., 2015*), ovaries (*Fichelson et al., 2009*; *Morris and Spradling, 2011*; *Reilein et al., 2018*), testis (*Cheng and Hunt, 2009*) and, more recently, adult midguts (*Deng et al., 2015*; *Xu et al., 2017*; *He et al., 2018*; *Martin et al., 2018*; *Hu and Jasper, 2019*). The small size of fruit flies makes it possible to culture whole intact organs.

However, in contrast to mammalian tissues, many of which are easily cultured for long periods, most *Drosophila* organ cultures are limited in time to less than a day. This reflects an incomplete understanding of the culture conditions required to fully sustain explanted *Drosophila* tissues. Current approaches for the live-imaging of the fly midgut are limited to 16 hr of imaging due to the poor survival of explanted tissues (*Martin et al., 2018*; *Hu and Jasper, 2019*). Moreover, temperature-sensitive gene expression, knock-down, and knock outs, some of *Drosophila* genetics strongest assets, cannot currently be implemented in combination with extended live-imaging because the elevated temperature further limits tissue viability (*Martin et al., 2018*).

To address these limitations, we developed an improved ex vivo culture system for the live-imaging of adult *Drosophila* midguts. Our culture system employs a novel tissue culture medium tailored to the needs of adult *Drosophila* cells and organs, and culture at an air-media interface to ensure optimal oxygenation. The technique has a straightforward design, allowing multiple samples to be prepared quickly and reproducibly. The setup allows the researcher to image up to 12 midguts simultaneously during live-imaging sessions of 48–72 hr. As the guts are fully explanted from the animal, every region of the organ is clearly available for imaging, thus expanding the number of questions that can be addressed. We show that, while in healthy explanted intestines progenitor cells are quiescent, midguts can still respond to damage ex vivo, with progenitors proliferating and differentiating in response to tissue damage. Our protocol can also be used in conjunction with temperature-sensitive gene expression or knock-down. We demonstrate this by genetically driving progenitor cell proliferation. Moreover, due to the extended live-imaging window our protocol allows, we were able to follow cells undergoing multiple rounds of mitosis. By combining a long 48–72 hr imaging window and the possibility to use advanced *Drosophila* genetic tools, we provide a useful tool to probe and understand the biology of epithelial tissues.

## Results

### A system for the long-term culture of adult *Drosophila* midguts

Current live-imaging protocols for the adult *Drosophila* midgut are limited to 16 hr imaging sessions (*Deng et al., 2015*; *Xu et al., 2017*; *He et al., 2018*; *Martin et al., 2018*; *Hu and Jasper, 2019*). To extend the survival of midguts ex vivo we developed a novel culture setup. The method is based on common available techniques for the culture of adult organs (*Sato et al., 2009*; *Handke et al., 2014*; *Reilein et al., 2018*; *Hu and Jasper, 2019*) but it includes a refinement of several steps: (1) the dissection procedure was optimized to reduce tissue damage; (2) explanted organs are cultured in a sandwiched structure of agarose, rather than in a dome; (3) midguts are placed at a liquid-air interface for improved oxygenation; (4) the culture media has been adjusted to better approximate adult hemolymph. Please see the Materials and methods section step-by-step descriptions of the procedure.

We found that the dissection technique is a key parameter in extending the viability of explanted midguts. Indeed, any stress (e.g. pulling and thus stretching the gut, pinching, etc...) introduced during dissection results in structural damage which lead to breaks of the epithelium during prolonged

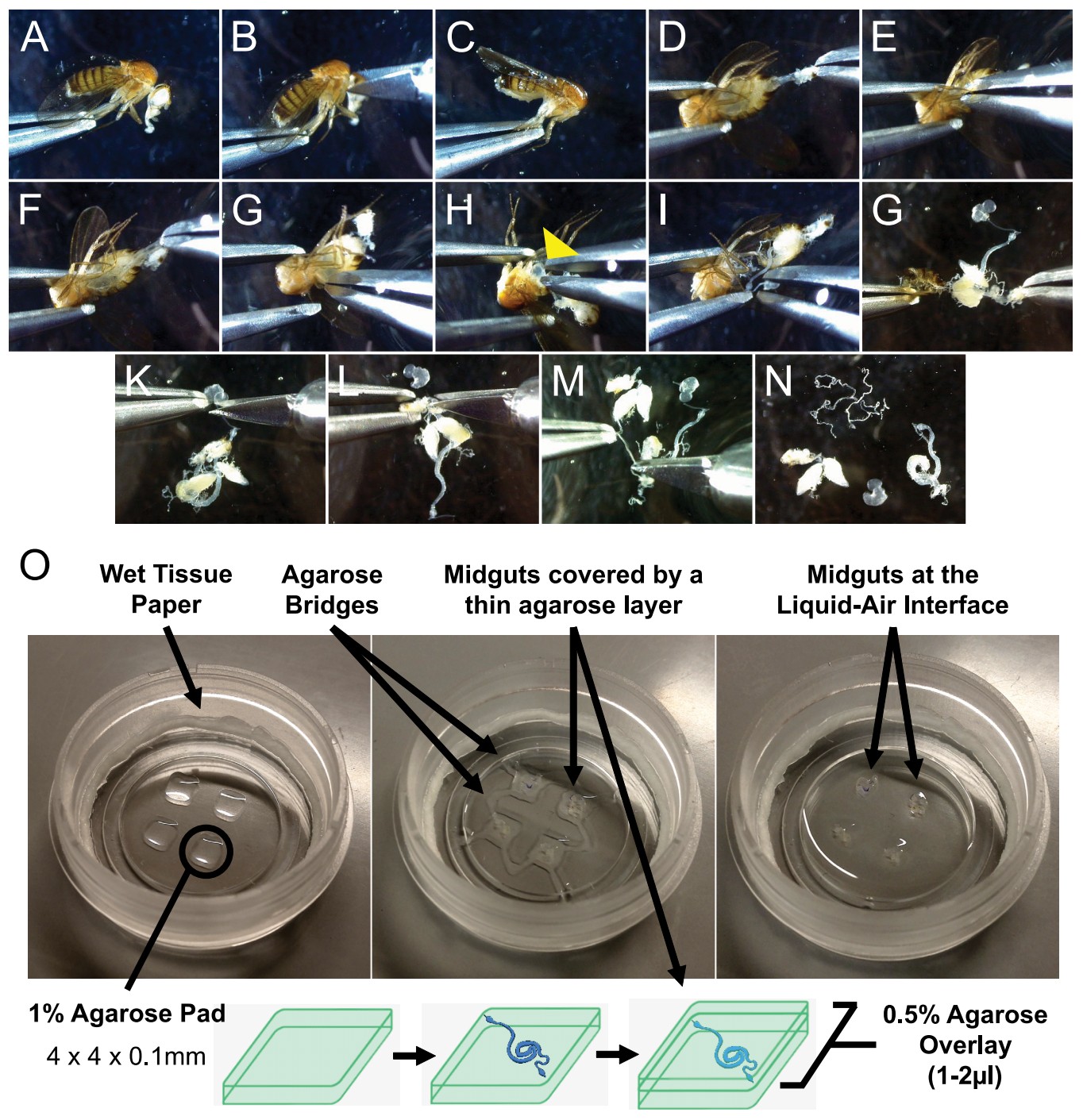

**Figure 1.** Sample preparation for live-imaging. (**A–N**) Minimal handling dissection used to gently explant adult *Drosophila* midguts, limiting the risk of damaging the intestines. See also *Figure 1—video 1*. (**O**) Culture chamber setup (left) and mounting of explanted midguts (middle and bottom diagram) to produce an air-liquid interface culture (right). See Materials and methods section and *Figure 1—video 1* for in-depth description.

The online version of this article includes the following video for figure 1:

**Figure 1—video 1.** Dissection technique to minimize damage during midgut explantation.

https://elifesciences.org/articles/76010/figures#fig1video1

culture. Hence, we have optimized the dissection procedure to limit the handling of the midgut and thus the risk of damaging it (*Figure 1A–N* and *Figure 1—video 1*).

A key element of our system is enveloping explanted organs in a sandwich of low-melt agarose (*Figure 1O*). Immobilization in gels is a common solution for ex vivo culture of organs and tissues, and is especially important to provide stability for live-imaging applications (*Sato et al., 2009*; *Handke et al., 2014*; *Reilein et al., 2018*; *Hu and Jasper, 2019*). For the adult *Drosophila* midgut specifically, an agarose gel also minimizes the effects of peristaltic movements, which, if uncontrolled, will impair imaging and can lead to epithelial tearing. Our approach is a slight departure from previously published techniques for *Drosophila* tissues (*Handke et al., 2014*; *Reilein et al., 2018*; *Hu and Jasper, 2019*) in that explanted midguts are transferred to evenly spaced thin agarose pads and then covered with an additional layer of agarose (*Figure 1O*). This sandwiched structure allows the guts to be held in place for imaging, while also protecting them from damage that can result from their contact with the culture chamber walls if left freely floating. The agarose pads can be reproducibly cast and are thin enough (~100 µm) not to interfere with imaging. Moreover, each agarose pad can comfortably house up to 3 midguts, allowing the multiple imaging of several explanted intestines. The agarose pads have the additional function of elevating the midguts from the bottom of the imaging chamber so that the surface of the agarose structure is directly exposed to air, creating an air-liquid interface. This is a key design element of the setup, as proper oxygenation was found to be essential for the long-term survival of explanted midguts (data not shown), similarly to what had been previously observed for the culture of wing imaginal discs (*Strassburger et al., 2017*). Moreover, during dissection the trachea surrounding the intestine are severed and can no longer oxygenate the intestinal epithelium.

To increase the stability of the setup, agarose bridges connect each agarose sandwich (*Figure 1O*, middle panel), allowing the sample to endure 3 days of continuous imaging (*Figure 2C* and *Figure 2—video 1*). Moreover, using a microscope equipped with an incubation chamber to control evaporation removes the need to replenish the imaging vessel with fresh culture medium. As such, the culture system, despite its simple design, is highly efficient and well suited for long-term imaging experiments.

## A culture medium tailored to adult *Drosophila* tissues enhances the survival of explanted midguts

One of the factors currently limiting the extended survival of explanted adult *Drosophila* tissues is the lack of culture media specifically designed for this task. To obviate this issue, we analyzed several parameters that distinguish the hemolymph of larvae, on which most current *Drosophila* cell culture media are based, to that of adult flies. We therefore compared the performance of different media formulations using the incorporation ex vivo of the cell-impermeable dye NucGreen (Thermofisher) as a measure of cell death (*Figure 2A*). The dye was added at the start of the culture, and its accumulation in the tissue measured over the course of three days. At the start of the culture, we usually observed NucGreen incorporation only in trachea, which are inevitably severed and damaged as a result of the dissection. We found that raising the concentration of trehalose, which is found at high levels in *Drosophila* circulation (*Pasco and Léopold, 2012*; *Dus et al., 2013*; *Park et al., 2014*; *Tennessen et al., 2014*; *Matsushita and Nishimura, 2014*), and mimicking the $Na^+/K^+$ ratio of adult hemolymph (*Singleton and Woodruff, 1994*; *Naikkhwah and O'Donnell, 2014*; *MacMillan et al., 2015b*; *MacMillan et al., 2015a*; *Olsson et al., 2016*) is sufficient to significantly reduce cell death after 24 hr of culture. As Schneider's medium (*Schneider, 1964*; *Schneider, 1966*; *Schneider, 1972*; *Schneider and Blumenthal, 1978*) is a common solution for several published protocols for the live-imaging of adult *Drosophila* tissues (*Fichelson et al., 2009*; *Reilein et al., 2018*; *Cheng and Hunt, 2009*; *Xu et al., 2017*; *Martin et al., 2018*), we modified it to raise the trehalose concentration to 50 mM and $Na^+/K^+$ ratio to levels similar to those found in adult *Drosophila* hemolymph (*Table 1*; *Dus et al., 2013*; *Park et al., 2014*; *Tennessen et al., 2014*; *Singleton and Woodruff, 1994*; *Naikkhwah and O'Donnell, 2014*; *MacMillan et al., 2015b*; *MacMillan et al., 2015a*; *Olsson et al., 2016*). Our tests showed that supplementing the culture medium with 10% fetal bovine serum (FBS) did not have a significant beneficial effect on the midgut epithelium (*Figure 2A*), but we did notice that FBS addition resulted in the visceral muscle remaining intact and capable of peristalsis for longer (data not shown). Not surprisingly (*Davis and Shearn, 1977*; *Britton and Edgar, 1998*), co-culturing explanted midguts with ovaries and abdomens lined with fat body (adipocytes) could also decrease cell death ex vivo. This effect was not due to the sequestration of NucGreen by ovaries and adipocytes as the

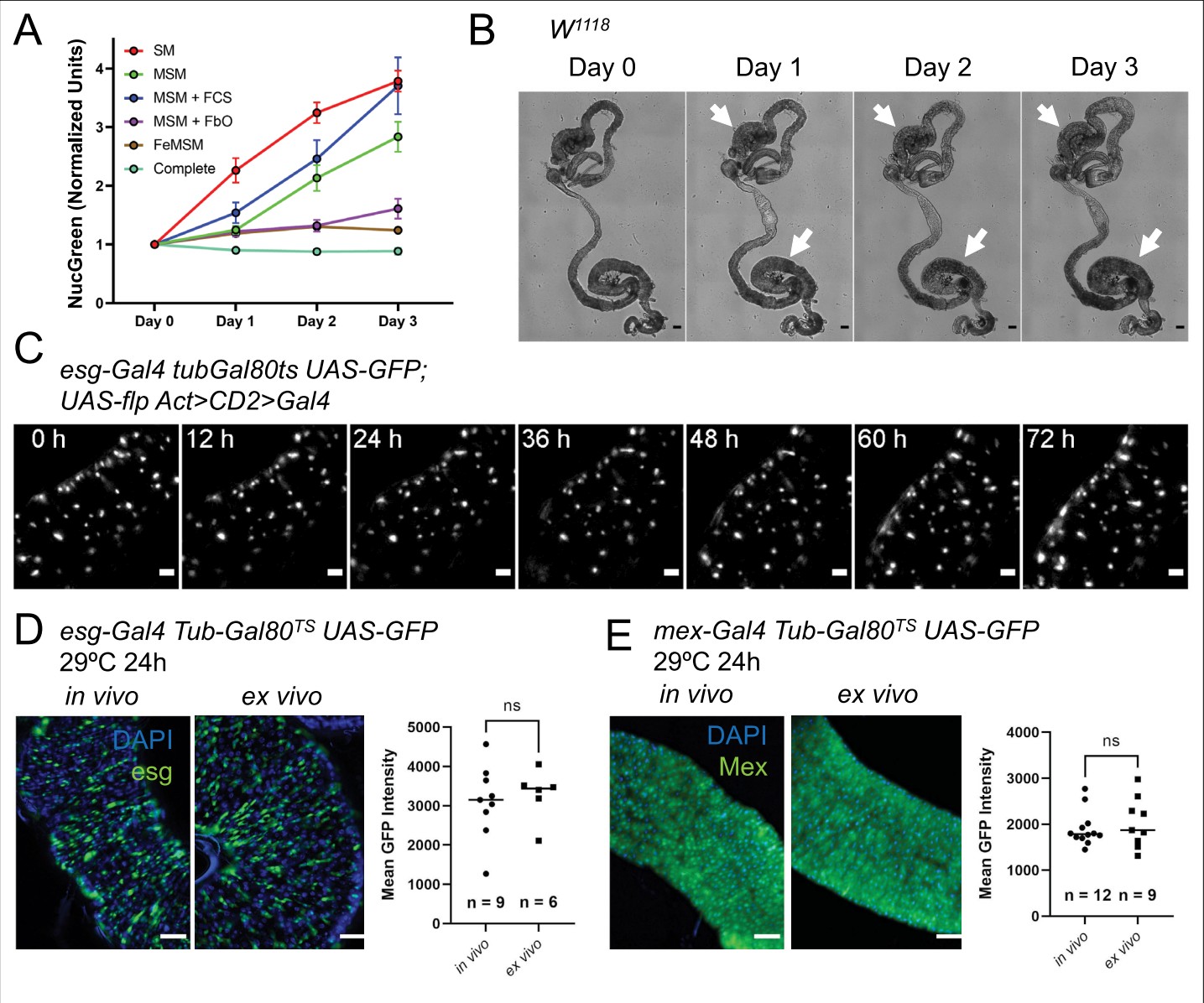

**Figure 2.** A custom culture medium sustains the midgut ex vivo. (**A**) Incorporation of the cell-impermeable dye NucGreen shows the levels of midgut cell death ex vivo in midguts cultured in: standard Schneider's medium (SM), modified Schneider's medium (MSM), MSM with 10% added fetal calf serum (FCS), MSM including co-culture with fat bodies and ovaries (FbO), culture in fly extract prepared in MSM (FeMSM), and the combination of all these conditions (Complete). Error bars represent the standard error of the mean. (**B**) Explanted midguts maintain their shape and tissue integrity for 3 days in culture. However, commensal bacteria keep growing in the lumen, especially in the posterior section where this can be seen as a darkening of the lumen (white arrows). Scale bar is 100 μm. (**C**) Maximum intensity projection of intestine expressing a lineage-tracing system under $esg^{TS}$ driver and imaged for 72 hr at 29 °C. The fly of origin was incubated at 29 °C for 24 hr prior to dissection. The intestine was undamaged during the course of imaging and did not show proliferation events. Scale bar is 5 μm. See also *Figure 2—video 1*. (**D–E**) Temperature-sensitive gene induction ex vivo is possible in both progenitor cells (**D**) and enterocytes (**E**). GFP levels expressed after 24 hr of gene induction are comparable between in vivo and ex vivo intestines. Explanted intestines were shifted to 29 °C immediately after sample preparation. In vivo controls were shifted simultaneously. Images are representative maximum intensity projections of posterior midguts. Scale bar is 50 μm. Each dot in the graphs represents the average GFP expression level in the posterior section of an intestine (T test). (ns, not significant).

The online version of this article includes the following video, source data, and figure supplement(s) for figure 2:

**Source data 1.** Raw data for *Figure 2A, D and E*.

**Figure supplement 1.** Ex vivo temperature sensitive gene expression of GFP.

**Figure 2—video 1.** 72 h live-imaging of an undamaged intestine expressing a lineage tracing system under esg^{TS} driver.

https://elifesciences.org/articles/76010/figures#fig2video1

**Table 1.** Modified Schneider's medium formulation.

Components whose concentration was modified are highlighted in yellow.

| Component | Schneider's medium | Modified Schneider's medium |
| --- | --- | --- |
| Amino Acids | Concentration (mM) | Concentration (mM) |
| Glycine | 3.3 | 3.3 |
| L-Arginine | 2.3 | 2.3 |
| L-Aspartic acid | 3.0 | 3.0 |
| L-Cysteine | 0.5 | 0.5 |
| L-Cystine | 0.4 | 0.4 |
| L-Glutamic Acid | 5.4 | 5.4 |
| L-Glutamine | 12.3 | 12.3 |
| L-Histidine | 2.6 | 2.6 |
| L-Isoleucine | 1.1 | 1.1 |
| L-Leucine | 1.1 | 1.1 |
| L-Lysine hydrochloride | 9.0 | 9.0 |
| L-Methionine | 5.4 | 5.4 |
| L-Phenylalanine | 0.9 | 0.9 |
| L-Proline | 14.8 | 14.8 |
| L-Serine | 2.4 | 2.4 |
| L-Threonine | 2.9 | 2.9 |
| L-Tryptophan | 0.5 | 0.5 |
| L-Tyrosine | 2.8 | 2.8 |
| L-Valine | 2.6 | 2.6 |
| beta-Alanine | 5.6 | 5.6 |
| Inorganic Salts | | |
| Calcium Chloride (CaCl2) (anhyd.) | 5.4 | 5.4 |
| MgCl2 Hexahydrate | 0.0 | 0.0 |
| Magnesium Sulfate (MgSO4) (anhyd.) | 15.1 | 15.1 |
| Potassium Chloride (KCl) | 21.3 | 21.3 |
| Potassium Phosphate monobasic (KH2PO4) | 3.3 | 3.3 |
| Sodium Bicarbonate (NaHCO3) | 4.8 | 4.8 |
| Sodium Chloride (NaCl) | 36.2 | 91.2 |

*Table 1 continued on next page*

*Table 1 continued*

| Component | Schneider's medium | Modified Schneider's medium |
| --- | --- | --- |
| Sodium Phosphate dibasic (Na2HPO4) anhydrous | 4.9 | 4.9 |
| Other Components | | |
| N-Acetyl Cysteine | 0.0 | 2.0 |
| Trisodium Citrate Dihydrate | 0.0 | 1.0 |
| Alpha-Ketoglutaric acid | 1.4 | 1.4 |
| D-Glucose (Dextrose) | 11.1 | 11.1 |
| Fumaric acid | 0.9 | 0.9 |
| Malic acid | 0.7 | 0.7 |
| Succinic acid | 0.8 | 0.8 |
| Trehalose | 5.8 | 55.8 |
| Yeastolate (g/l) | 2000.0 | 2000.0 |

dye was supplemented at a saturating concentration and its incorporation in these organs/tissues was mostly limited to areas mechanically damaged during dissection. As ovaries and fat bodies may enhance the survival of explanted midguts by secreting growth factors and/or nutrients, we reasoned that fly extract might have a similar effect. Indeed, midguts cultured in 100% fly extract prepared using modified Schneider's medium had greatly reduced rates of cell death

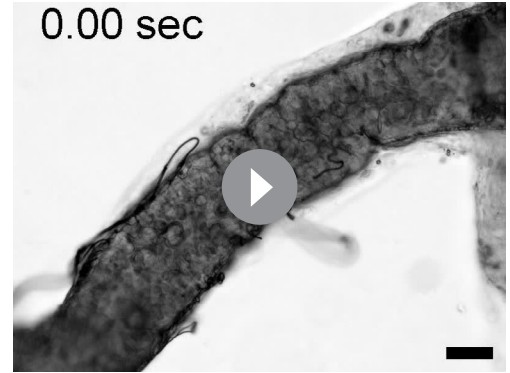

**Video 1.** Example of visceral muscle peristaltic movements after 3 days ex vivo. Intestine was cultured in the presence of the calcium blocker isradipine. After 2-3 days in culture, the effect of the drug dissipates and the visceral muscle restarts its regular peristaltic movements. Scale bar is 50µm.

https://elifesciences.org/articles/76010/figures#video1

over time (*Figure 2A*). The calcium blocker isradipine was also added to the medium to reduce peristaltic movements, improving midgut survival and imaging. This drug did not have a noticeable long-term effect on visceral muscle viability. Indeed, its effect appeared to wear off after 48–72 hr, at which point the visceral muscle resumed regular peristalsis (*Video 1*). Lastly, as we designed the medium for the purpose of extended live-imaging, we also added N-acetyl cysteine and sodium citrate as antioxidant agents to reduce phototoxicity (*Icha et al., 2017*). The former is an antioxidant widely used in cell culture (*Ezeriņa et al., 2018*). Citrate is also known to have antioxidant properties (*Wu et al., 2019*) and to be present in *Drosophila* hemolymph at detectable levels (*Echalier et al., 2018*). Combining all the findings mentioned above resulted in culture conditions (see Materials and methods) that minimized cell death and allowed the live-imaging of explanted midguts for up to 3 days (*Figure 2C* and *Figure 2—video 1*), a significant improvement over previously published culture protocols for the midgut in which imaging was limited to 16 hr (*Martin et al., 2018*; *Hu and Jasper, 2019*).

We successfully imaged cultured intestines with inverted microscopes of different kinds, including widefield (Nikon TiE), scanning confocal (Leica SP8), and lattice lightsheet (Zeiss LLS7). We did not notice signs of phototoxicity with any of these microscope types in samples imaged for 48 hr, but acquisition parameters had to be chosen with care. For example, by optimizing laser power and dwell time, single intestines could be imaged at high frame-rates (15 min) without noticeable phototoxicity for at least 48 hr even with a confocal microscope (Video 5). However, we found confocal imaging to be less optimal than widefield microscopy for high frame-rate imaging due to the longer acquisition time required for multiple midguts in a single sample.

In examining cultured midguts, we observed the accumulation of luminal contents (*i.e.* previously ingested food) in the posterior midgut (*Figure 2B*, white arrows). This appeared to be caused by peristaltic movements of the visceral muscle which persisted despite the use of the calcium blocker isradipine, although this drug suppressed them significantly. These areas were found to darken and expand over the course of culture, an effect we attribute to the growth of enteric bacteria, which eventually caused cell death and tissue damage. We found that feeding flies fresh food in the days prior to an experiment and a short (~3–6 hr) starvation prior to dissection reduced the negative effect of food accumulation and growth of enteric bacteria. Supplementing the culture medium with antibiotics also enhanced explanted midgut survival. Lastly, it is reasonable to assume that axenically reared flies should be immune to the issue of growing enteric bacteria.

## Transgenic gene expression in midgut explant culture

One of the most striking features of *Drosophila melanogaster* as a research model is the wide range of readily available genetic tools, allowing the cell-type-specific and temporally-controlled activation or suppression of expression of genes of interest. The possibility to use such genetic tools in a live-imaging setup is highly advantageous, but so far the temporal control of gene expression ex vivo has proven unfeasible (*Martin et al., 2018*). To further assay the behavior of midgut cells in explanted organs, we tested the induction of UAS-GFP by cell type-specific, temperature-sensitive Gal4 drivers (*Figure 2D–E*; *Brand and Perrimon, 2018*; *McGuire et al., 2003*). As expected, midguts explanted from flies grown at the restrictive temperature (18 °C) did not show any GFP expression (*Figure 4—video 1* and *Figure 2—figure supplement 1*, left panels). However, when incubated at the permissive temperature (29 °C) at 0 hr after explant (*Figure 4—video 1* and *Figure 2—figure supplement 1*, right panels), they started expressing the UAS-GFP transgene. Moreover, GFP expression could also be induced in intestines cultured at 18 °C for 24 hr and then shifted to the permissive temperature (29 °C), indicating that the epithelium remains viable and genetically functional long-term (*Figure 2—figure supplement 1B*). Indeed, the GFP intensity in midguts cultured at 29 °C for 24 hr from the time of dissection was similar between in vivo and ex vivo conditions for both progenitor cells (*Figure 2D*) and enterocytes (*Figure 2E*) using the *escargot*- (*esg*) and *mex*-Gal4 Gal80$^{TS}$ driver lines, respectively. This indicates that transcription and protein synthesis are maintained at normal levels in our culture system and shows that this system can be used to assay the effects of transgene induction in real time. Interestingly, progenitor cells were found to be asynchronous in their expression of the reporter GFP (Figure 4Aand *Figure 4—video 1*). Usually, the first GFP + cells were detected at 8–10 hr after temperature shift, though some cells achieved detectable GFP levels only 20 hr after that. Overall, if the shift to 29 °C coincided with the start of imaging, all progenitor cells usually expressed detectable levels of GFP by the 36 hr time-point. This could be explained by variations in the activity of the *esg*

promoter or by different global rates of transcription and translation, which in turn could be indicative of different cell states.

## Progenitor cells require stimulation to proliferate and differentiate

When imaging explanted adult *Drosophila* midguts we observed *esg*-expressing progenitor cells (ISCs and EBS) to be quiescent (*Figure 3A* and *Figure 2—video 1*, *Figure 3—video 1*, *Figure 3—video 4*). Progenitor cells in explanted healthy guts did not show major changes in their GFP levels, indicative of continuous *esg* expression, nor in their nuclear size (*Figure 3D, F and G* and *Figure 3—video 4*), which indicates that DNA content remains constant. Via cell tracking we also observed that GFP-labeled progenitor cells did not divide in healthy explanted intestines (*Figure 4*). It is important to note that in said healthy intestines we did not observe cell death or enterocyte (EC) extrusion events until after 48–72 hr of culture. As healthy enterocytes are known to suppress ISC proliferation (*Liang et al., 2017*), this may explain the lack of cell division in our explants. The suppression of peristaltic movements by isradipine may also prevent EC loss by reducing mechanical tissue stress (*Li et al., 2018*).

A key feature of *Drosophila* intestinal progenitor cells is their ability to rapidly respond to tissue damage (*Ohlstein and Spradling, 2006*; *Micchelli and Perrimon, 2006*; *Jiang et al., 2009*; *Patel et al., 2019*; *Amcheslavsky et al., 2009*). To confirm that this capability is maintained ex vivo, we fed female flies Sodium dodecyl sulfate (SDS) mixed with solid fly food at a final concentration of 0.2% (v/w) for 18 hr. Following this treatment, the flies were fed 0.05% sucrose in aqueous solution on a cotton pad for an additional 4–6 hr prior to dissection, in order to flush the SDS from the gut. This protocol caused only mild initial tissue damage, and was advantageous as it did not result in the retention of high levels of SDS in explanted guts. However, by mixing the SDS-laced food with a blue food-safe dye, we found that some low amounts of SDS-laced food do persist in the lumen of intestines at the time of dissection. The transient exposure to SDS and the low amounts retained in the lumen resulted in gradual EC death and/or extrusion, allowing the live-imaging of damage response. When imaged, intestines from SDS-fed flies showed progenitor cells that become highly motile, similar to what was recently observed in similarly damaged intestines (*Hu et al., 2021*) . ISC proliferation events could also be detected (*Figure 3C*). However, ISC mitoses could generally be observed only in cases where tissue damage was extensive, as evidenced by the appearance of pyknotic or fragmented nuclei and the extrusion of multiple (>3) contiguous ECs. As this tended to happen towards late time-points after dissection, only a few (1-2) mitotic events per imaged field could be observed with this damage protocol. In addition to ISC divisions, many progenitor cells could be observed growing in nuclear size while simultaneously losing GFP expression (*Figure 3B and E–G* and *Figure 3—video 3*, *Figure 3—video 4*) indicative of differentiation events towards the EC identity. Indeed, differentiating progenitor cells could be seen replacing dying enterocytes (*Figure 3—video 3*).

We then tested whether these observations could be made in an in vivo condition. Before dissection, flies were fed either a control diet or SDS-laced food as described above. Flies dissected at later time-points (i.e. 24 and 48 h after the initial SDS feeding) where kept on fresh food, thus allowing the intestine to clear any trace of SDS and recover. This is different to the ex vivo condition, where a small amount of SDS is retained in the lumen and continues to damage the intestine during imaging. Although we found it impossible and impractical to perfectly match the luminal SDS content in vivo and ex vivo, the in vivo condition did confirm our ex vivo observations (*Figure 3—figure supplement 1*). After SDS feeding, we observed several esg + cells with larger nuclei (*Figure 3—figure supplement 1A-B*). At later time-points many differentiating cells with large nuclei and low GFP expression were also present, similarl to what was observed ex vivo. Moreover, similarly to explanted midguts, proliferative cells could be found almost exclusively at 48 hr after the end of the SDS treatment (*Figure 3—figure supplement 1C*).

A limitation of the SDS-feeding protocol described above is that midguts were damaged prior to imaging. Therefore, it cannot be excluded that the observed differentiation events were EBs that were poised to differentiate prior to SDS feeding, and then differentiated due to the damage stimulus. To confirm that, at the time of damage, pre-existing EBs were indeed capable of responding to tissue stress, we used a thin tungsten needle to create a small lesion in explanted intestines. The lesions perforated both visceral muscle and epithelial layers, but left the peritrophic matrix intact (*Figure 3—figure supplement 2A*). Intestines were then enveloped in agarose and imaged within

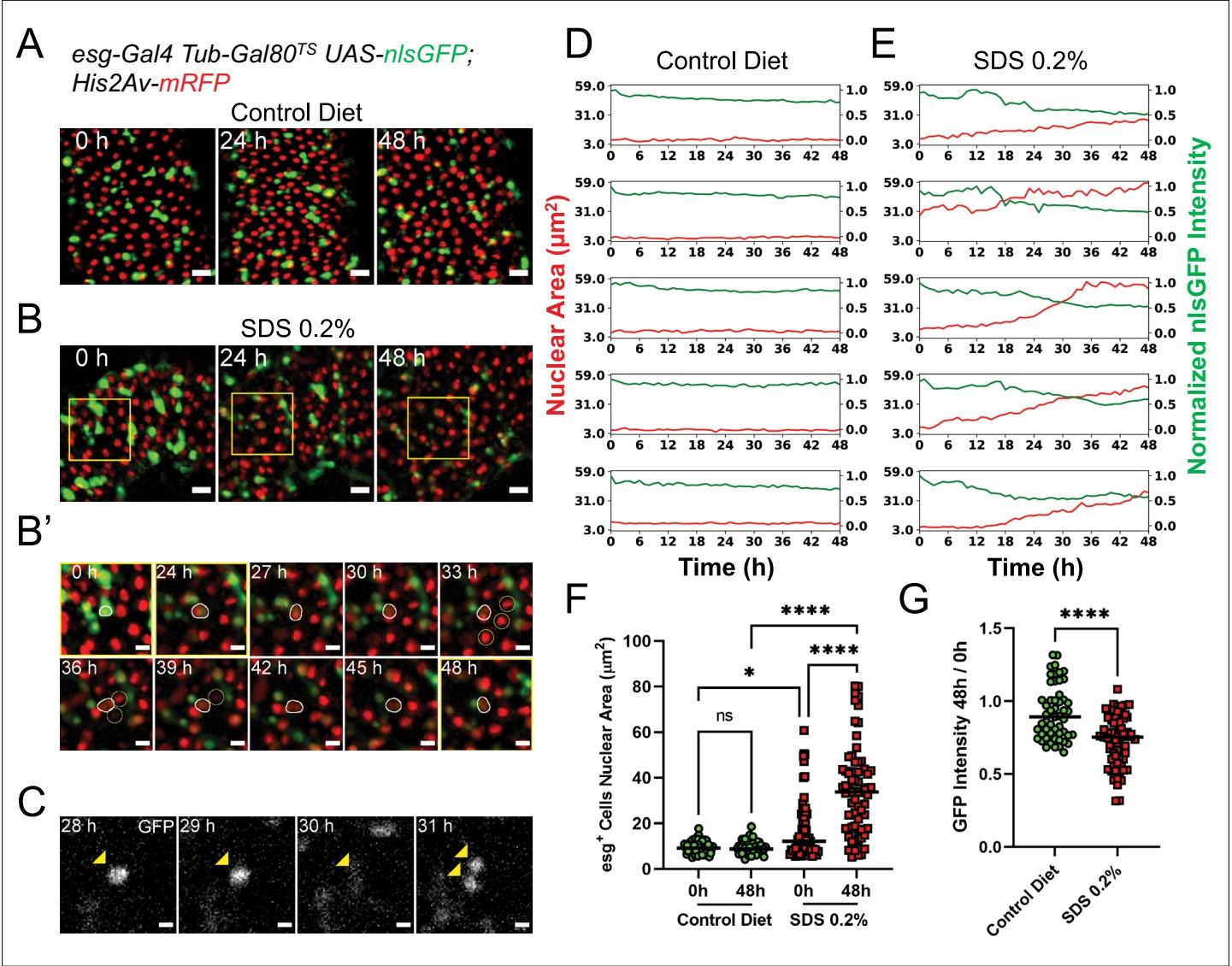

**Figure 3.** Midguts respond to damage ex vivo. (**A**) Healthy gut fed a control diet showing no sign of tissue damage. Images are maximum intensity projections. See also *Figure 3—video 1*. Scale bar is 20 μm. (**B**) Gut from fly fed SDS 0.2% overnight showing progressive tissue repair ex vivo mediated by progenitor cells proliferation and differentiation. As cells differentiate, nlsGFP expression is gradually lost. Area delimited by the yellow rectangle is enlarged in (**B'**). Images are maximum intensity projections. See also *Figure 3—video 2*. Scale bar is 20 μm. (**B'**) Enlargement of panel (**B**) showing an *esg*+ cell (white outline) differentiating and replacing dying enterocytes (yellow circles). As the cell differentiates, nlsGFP expression is lost and its nucleus grows larger. Images are maximum intensity projections. See also *Figure 3—video 3*. Scale bar is 10 μm. (**C**) Example of stem cell dividing upon tissue damage initiated by SDS feeding. Cell is marked by the expression of nlsGFP (yellow arrowhead). Note that in the 30 hr time-point the cell is in mitosis and so the nlsGFP signal is mostly lost due to nuclear envelope breakdown. After mitosis, the nuclear envelope is reformed and GFP re-accumulates in the nucleus of the daughter cells. Images are single z-slices. Scale bar is 5 μm. (**D**) Plots of nuclear area (red) and mean nuclear GFP intensity (green) from single progenitor cells from healthy guts. Note that both nuclear size and GFP signal, despite a small gradual dip caused by photobleaching, are stable for the duration of the imaging session. See also *Figure 3—video 4*, left column. (**E**) Plots of nuclear area (red) and mean nuclear GFP intensity (green) in single progenitor cells from SDS-damaged guts. Note that nuclear size increases, while GFP signals dims over time, suggesting EB to EC differentiation. See also *Figure 3—video 4*, right column. (**F**) Quantification of nuclear area of esg+ cells in control (green) and SDS-damaged (red) intestines cultured ex vivo. Each cell was measured at imaging start (i.e. 0 hr) and 48 hr later. In control guts, esg+ cells remain quiescent and have no change in nuclear size. Due to the SDS treatment, several progenitor cells have a large nucleus already at the time of dissection. Furthermore, most progenitor cells' nuclear area significantly increases during the course of imaging (Two-way Anova and Šídák's multiple comparisons test). (**G**) Ratio of nuclear GFP intensity for individual esg+ cells from control or SDS-damaged intestines cultured ex vivo. While cells in control intestines do not lose GFP expression (except for a minor loss due to photobleaching), several cells in the SDS-treated intestines show a significant loss of GFP intensity, suggesting their differentiation and change of cell identity (T test). (ns, not significant, *, p<0.05; ****, p<0.0001).

The online version of this article includes the following video, source data, and figure supplement(s) for figure 3:

*Figure 3 continued on next page*

*Figure 3 continued*

**Source data 1.** Raw data for *Figure 3D–G*.

**Figure supplement 1.** Damage response to SDS feeding in vivo.

**Figure supplement 1—source data 1.** Raw data for *Figure 3—figure supplement 1B-C*.

**Figure supplement 2.** Tissue repair in midguts damaged ex vivo.

**Figure supplement 3.** *Notch* knock-down induces tumorigenesis ex vivo.

**Figure 3—video 1.** 48h live-imaging of a healthy intestine expressing His2Av.mRFP (red) and esg$^{TS}$ -driven nlsGFP (Green) induced 24h prior to imaging.
https://elifesciences.org/articles/76010/figures#fig3video1

**Figure 3—video 2.** 48 hr live-imaging of an intestine from a SDS-fed fly expressing His2Av.mRFP (red) and esg$^{TS}$-driven nlsGFP (Green) induced 24 hr prior to imaging.
https://elifesciences.org/articles/76010/figures#fig3video2

**Figure 3—video 3.** Detailed view of *Figure 3—video 2* showing a progenitor cell (white outline) differentiating and replacing dying enterocytes (yellow circles) in an intestine from a SDS-fed fly expressing His2Av.mRFP (red) and esg$^{TS}$ -driven nlsGFP (Green) induced 24h prior to imaging.
https://elifesciences.org/articles/76010/figures#fig3video3

**Figure 3—video 4.** Examples of esg+ cells from control or SDS-damaged intestines expressing His2Av.mRFP (red) and esg$^{TS}$ -driven nlsGFP (Green) induced 24h prior to imaging.
https://elifesciences.org/articles/76010/figures#fig3video4

**Figure 3—video 5.** 20h live-imaging of an intestine expressing esg-driven nlsGFP (right) damaged by needle poke 10' prior to imaging.
https://elifesciences.org/articles/76010/figures#fig3video5

**Figure 3—video 6.** 24h live-imaging of an undamaged intestine expressing His2Av.mRFP (red) and upd3-driven GFP (green).
https://elifesciences.org/articles/76010/figures#fig3video6

**Figure 3—video 7.** 24h live-imaging of an intestine damaged using a tungsten needle and expressing His2Av.mRFP (red) and upd3-driven GFP (green).
https://elifesciences.org/articles/76010/figures#fig3video7

**Figure 3—video 8.** 48h live-imaging of an undamaged intestine expressing His2Av.mRFP (red) and esg$^{TS}$ -driven nlsGFP (Green) and N$^{RNAi}$ induced 24h prior to imaging.
https://elifesciences.org/articles/76010/figures#fig3video8

**Figure 3—video 9.** 48h live-imaging of a damaged intestine expressing His2Av.mRFP (red) and esg$^{TS}$ -driven nlsGFP (Green) and N$^{RNAi}$ induced 24h prior to imaging.
https://elifesciences.org/articles/76010/figures#fig3video9

20' of the time of damage. Similarly to the SDS damage protocol, we were able to record progenitor cells both dividing and differentiating (*Figure 3—figure supplement 2B, C* and *Figure 3—video 5*). Moreover, wounding stimulated the robust expression of the cytokine Unpaired 3 (Upd3), whose function in midgut damage response is well documented (*Figure 3—figure supplement 2D, E* and *Figure 3—video 6*; *Figure 3—video 7*; *Jiang et al., 2009*; *Patel et al., 2015*; *Buchon et al., 2009*).

All in all, our observations suggest that both ISCs and EBs are quiescent in undamaged intestines. While previous studies did show varying rates of ISC proliferation in healthy midguts in vivo (*Ohlstein and Spradling, 2006*; *Micchelli and Perrimon, 2006*; *Jiang et al., 2009*; *Reiff et al., 2019*; *de Navascués et al., 2012*), this may be attributed to cell death (*Liang et al., 2017*) as well as the passage of food through the intestinal lumen (*Li et al., 2018*), both of which do not occur in our midgut cultures until later time-points (i.e. after 48–72 hr). Moreover, several previous studies used lineage-tracing tools requiring a 37 °C heat shock for their activation, a treatment known to induce ISC proliferation (*Beebe et al., 2010*; *de Navascués et al., 2012*; *Lin et al., 2008*; *Shaw et al., 2010*; *Strand and Micchelli, 2011*). Indeed, in the presence of tissue damage, progenitor cells could be robustly activated in explanted intestines, indicating that both ISCs and EBs are capable of responding to their environment after organ explant.

## Loss of *Notch* drives tumorigenesis ex vivo

A key pathway necessary for progenitor cell differentiation is the *Delta*/Notch signaling that occurs between ISCs and adjacent EBs (*Ohlstein and Spradling, 2006*; *Micchelli and Perrimon, 2006*; *Ohlstein and Spradling, 2007*). The depletion of *Notch* (N) by RNA interference (N$^{RNAi}$) in esg$^+$ cells has been shown to promote the formation of undifferentiated tumor-like masses in vivo (*Ohlstein and Spradling, 2006*; *Micchelli and Perrimon, 2006*; *Patel et al., 2015*; *Ohlstein and Spradling, 2007*).

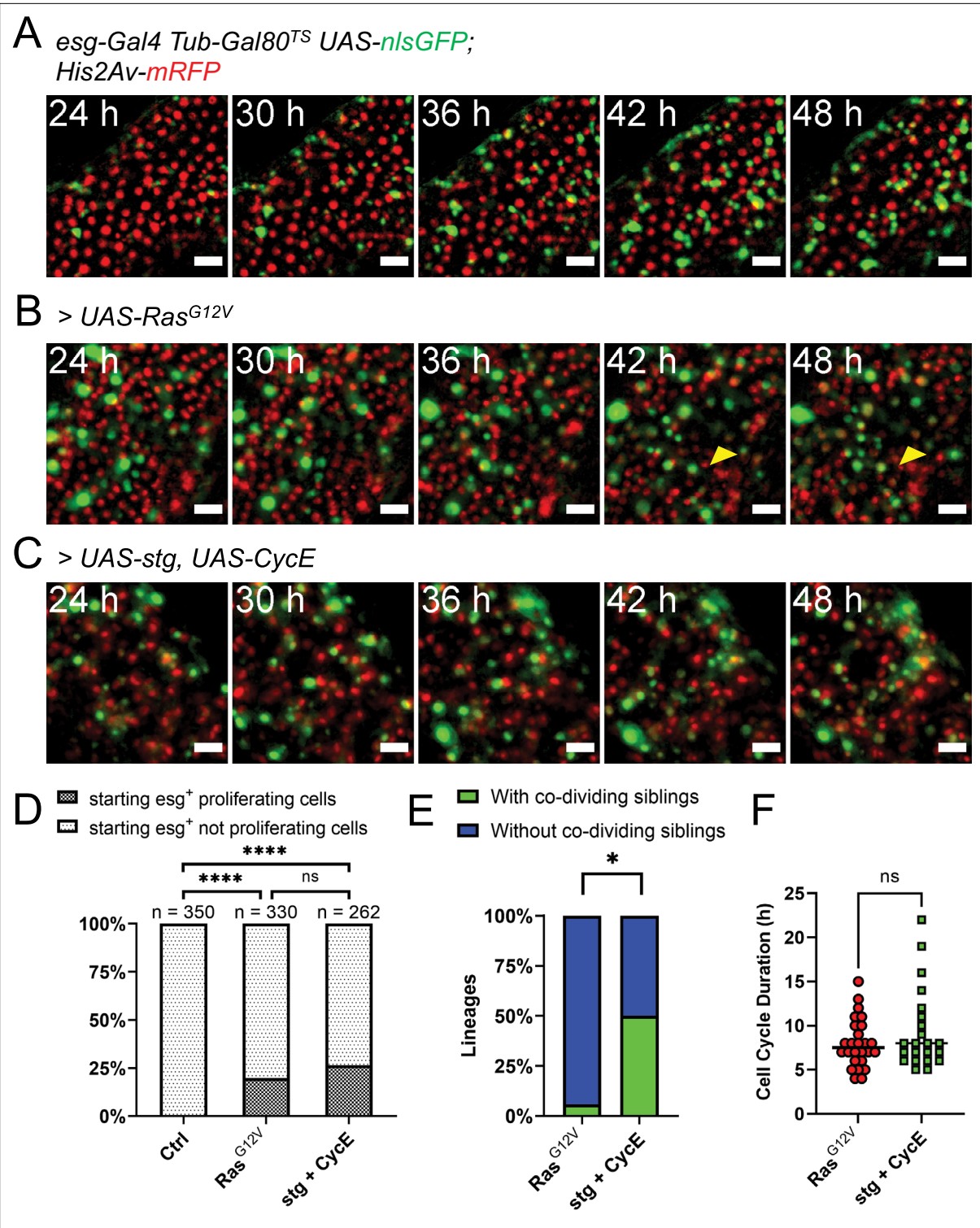

**Figure 4.** Genetic induction of intestinal stem cell proliferation. (**A**) Live-imaging of a normal intestine expressing nlsGFP in progenitor cells via *esg^TS* driver. Intestines were shifted to 29 °C at the start of imaging. Note the gradual accumulation of GFP signal showing variability between cells in terms of both time and intensity. Images are maximum intensity projections. See also *Figure 4—video 1*. Scale bar is 20 μm. (**B**) Live-imaging of an intestine expressing nlsGFP and *Ras^G12V* in progenitor cells via *esg^TS* driver. Intestines were shifted to 29 °C at the start of imaging. GFP+ cells can be seen rapidly dividing and displacing mature enterocytes (yellow arrowhead). Images are maximum intensity projections. See also *Figure 4—video 2*. Scale bar is 20 μm. (**C**) Live-imaging of an intestine expressing nlsGFP, *stg*, and *CycE* in progenitor cells via *esg^TS* driver. Intestines were shifted to 29 °C at the start of imaging. GFP+ cells can be seen rapidly dividing. Images are maximum intensity projections. See also *Figure 4—video 3*. Scale bar is 20 μm.

*Figure 4 continued on next page*

*Figure 4 continued*

(**D**) Quantification of observed proliferating and non-proliferating esg+ progenitor cells in control and intestines expressing *Ras^{G12V}* or stg and *CycE*. Only cells observed from the moment they expressed visible levels of GFP to the end of the imaging session are included. The progeny of observed mitoses were not counted for this analysis. (**E**) Frequency of stem cell lineages with divisions giving rise to two proliferative cells (green), compared between intestines expressing either *Ras^{G12V}* or stg and *CycE* (Fisher's exact test). (**F**) Quantification of cell cycle durations of progenitor cells expressing either *Ras^{G12V}* or stg and *CycE*, and nlsGFP. No significant difference was found between the two genotypes (Mann-Whitney test; ns, not significant; *, $p<0.05$).

The online version of this article includes the following video and source data for figure 4:

**Source data 1.** Raw data for *Figure 4D and F*.

**Figure 4—video 1.** 48h live-imaging of a healthy intestine expressing His2Av.mRFP (red) and esg^{TS}-driven nlsGFP (Green) induced after dissection.
https://elifesciences.org/articles/76010/figures#fig4video1

**Figure 4—video 2.** 48h live-imaging of an intestine expressing His2Av.mRFP (red) and esg^{TS}-driven nlsGFP (Green) and Ras^{G12V} induced after dissection.
https://elifesciences.org/articles/76010/figures#fig4video2

**Figure 4—video 3.** 48h live-imaging of an intestine expressing His2Av.mRFP (red) and esg^{TS}-driven nlsGFP (Green), stg, and CycE induced after dissection.
https://elifesciences.org/articles/76010/figures#fig4video3

Notably, tumor initiation by N-depleted ISCs requires proliferation induced by tissue stress (*Patel et al., 2015*; *Apidianakis et al., 2009*). Consistent with this, when we explanted midguts from flies expressing N^{RNAi} in progenitor cells for 24 h prior to dissection and cultured them for 48 hr, we did not observe tumorigenesis in the absence of tissue damage (*Figure 3—figure supplement 3A* and *Figure 3—video 8*). However, when guts were accidentally damaged during dissection (*Figure 3—figure supplement 3*, yellow ellipses), esg+ cells started to proliferate (*Figure 3—figure supplement 3B*, yellow squares, and *Figure 3—video 9*). Notably, progenitor cells remained small and did not lose GFP expression (compare *Figure 3B* and *Figure 3—figure supplement 3B* as well as *Figure 3—video 2* and *Figure 3—video 9*), suggesting a block of differentiation ex vivo in response to *Notch* knock-down.

## Intestinal progenitor proliferation can be genetically stimulated ex vivo

EGFR-Ras-Erk signaling is activated upon gut tissue damage, and this pathway is required for stem cell activation in the adult *Drosophila* intestine (*Buchon et al., 2010*; *Biteau and Jasper, 2011*; *Jiang et al., 2011*). Previous experiments showed that expression of a constitutively active form of *Ras* (*Ras^{G12V}*) strongly promotes stem cell proliferation in adult midguts (*Jiang et al., 2011*; *Jin et al., 2015*). As our culture protocol allows the expression of transgenes ex vivo, we induced *Ras^{G12V}* in explanted midguts using the *esg-Gal4 Gal80^{TS}* progenitor-specific driver gene combination (*esg^{TS}*). When explanted midguts were shifted to the permissive temperature (29 °C) at the start of imaging, progenitor cells, marked by GFP co-expressed with *Ras^{G12V}*, started to rapidly proliferate (*Figure 4B*). About 20% of progenitor cells tracked from the moment they expressed visible amounts of GFP till the end of the imaging session, were seen proliferating (*Figure 4D*). The progeny of observed mitoses were not counted for this analysis. However, since many cells were not trackable for the duration of the imaging session due to major tissue rearrangements, we may be underscoring the percentage of proliferative GFP+ cells. As GFP+ cells accumulated in the tissue, enterocytes were displaced and extruded from the epithelium (*Figure 4B*, yellow arrowhead, and *Figure 4—video 2*). Interestingly, some GFP+ progenitor cells did not proliferate, but their nuclei rapidly grew in size, which aligns with the previously reported role of EGFR signaling in promoting EB growth (*Jiang et al., 2011*; *Xiang et al., 2017*). Moreover, these rapidly growing progenitors also lost GFP expression during the course of imaging, which suggests their differentiation towards the EC lineage.

The extended live-imaging that our protocol allows also permitted us to follow cells through multiple rounds of mitosis. Using manual 3-dimensional (3D) cell tracking, we reconstructed the lineages of 17 dividing cells that expressed *Ras^{G12V}* (see *Figure 5* and *Figure 5—video 1* for an example). As ISCs are the only cell type in the *Drosophila* intestine that normally divide multiple times (*Ohlstein and Spradling, 2006*; *Micchelli and Perrimon, 2006*; *Chen et al., 2018*), the founding cells in these lineages were most likely ISCs. However, a recent work suggested that Ras^{G12V} can push a small number of EBs to de-differentiate to an ISC state (*Tian et al., 2021*). We directly measured the duration of progenitors' cell cycle in these lineages to be $8\pm2.76$ hr (*Figure 4F*). Interestingly, of 17

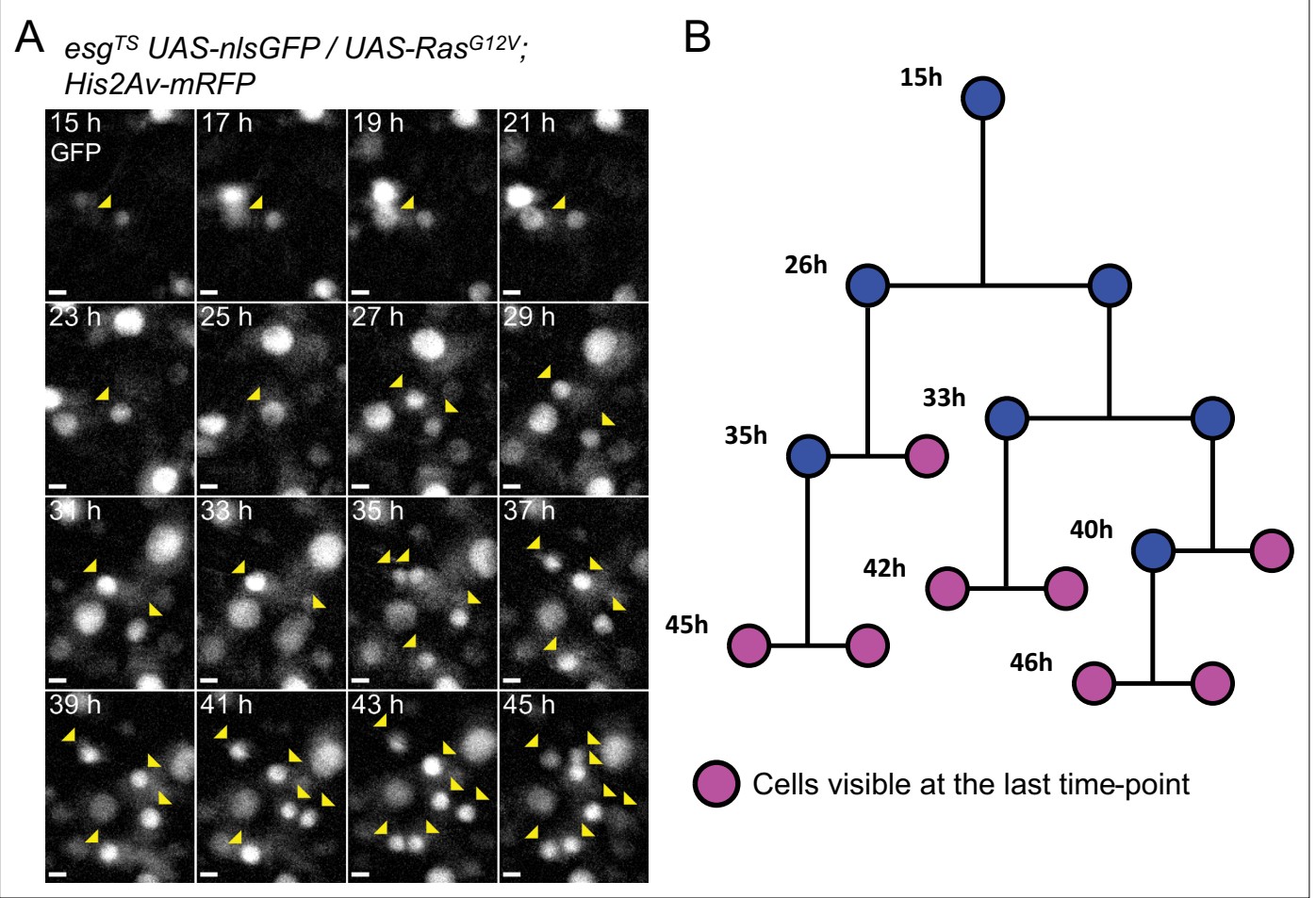

**Figure 5.** Example of a reconstructed ISC lineage. (**A**) Time-lapse of a single Ras$^{G12V}$-expressing ISC undergoing multiple rounds of mitoses. The nlsGFP channel is shown. Cells belonging to the lineage are marked by yellow arrowheads. (**B**) Lineage diagram. Cells visible at the last time-point shown are marked in magenta. Images are maximum intensity projections of z-slices encompassing the cells in the lineage. See also *Figure 5—video 1*. Scale bar is 5 μm.

The online version of this article includes the following video for figure 5:

**Figure 5—video 1.** :Example of ISC lineage.

https://elifesciences.org/articles/76010/figures#fig5video1

---

*Ras$^{G12V}$*-expressing ISC lineages characterized, one had divisions where both daughter cells from the first recorded division were seen dividing further (*Figure 4E*). For simplicity, pairs of dividing daughter cells will be henceforth be referred to as 'co-dividing siblings' (see *Figure 6A–B* and *Figure 6—video 1* for an example). These are most likely symmetric divisions that give rise to two new ISCs. However, since we did not track differentiation markers in our lineages, other possibilities cannot be excluded. For example, after being generated from an ISC division, enteroendocrine (EE) progenitors are also known to divide once to give rise to two mature EEs (*Chen et al., 2018*). Moreover, we cannot exclude that cells that were not observed to divide during our imaging session would not divide again, given enough time.

Since *Ras$^{G12V}$* stimulation results in multiple cellular changes, we also tested whether we could stimulate ISC proliferation more directly. For this we used the *esg$^{TS}$* driver to co-express *string* (*stg*), a *Cdc25C* homolog, and *Cyclin E* (*CycE*) (*Figure 4C* and *Figure 4—video 1*). String directly activates Cdk1 to promote mitosis, whereas CycE directly activates Cdk2 to promote DNA replication and S-phase progression. The combined expression of these two gene products is sufficient to strongly induce ISC proliferation (*Kohlmaier et al., 2015*). Notably, Stg and CycE have also been shown to promote EB mitoses (*Kohlmaier et al., 2015*), although much less strongly than in ISCs. The fraction

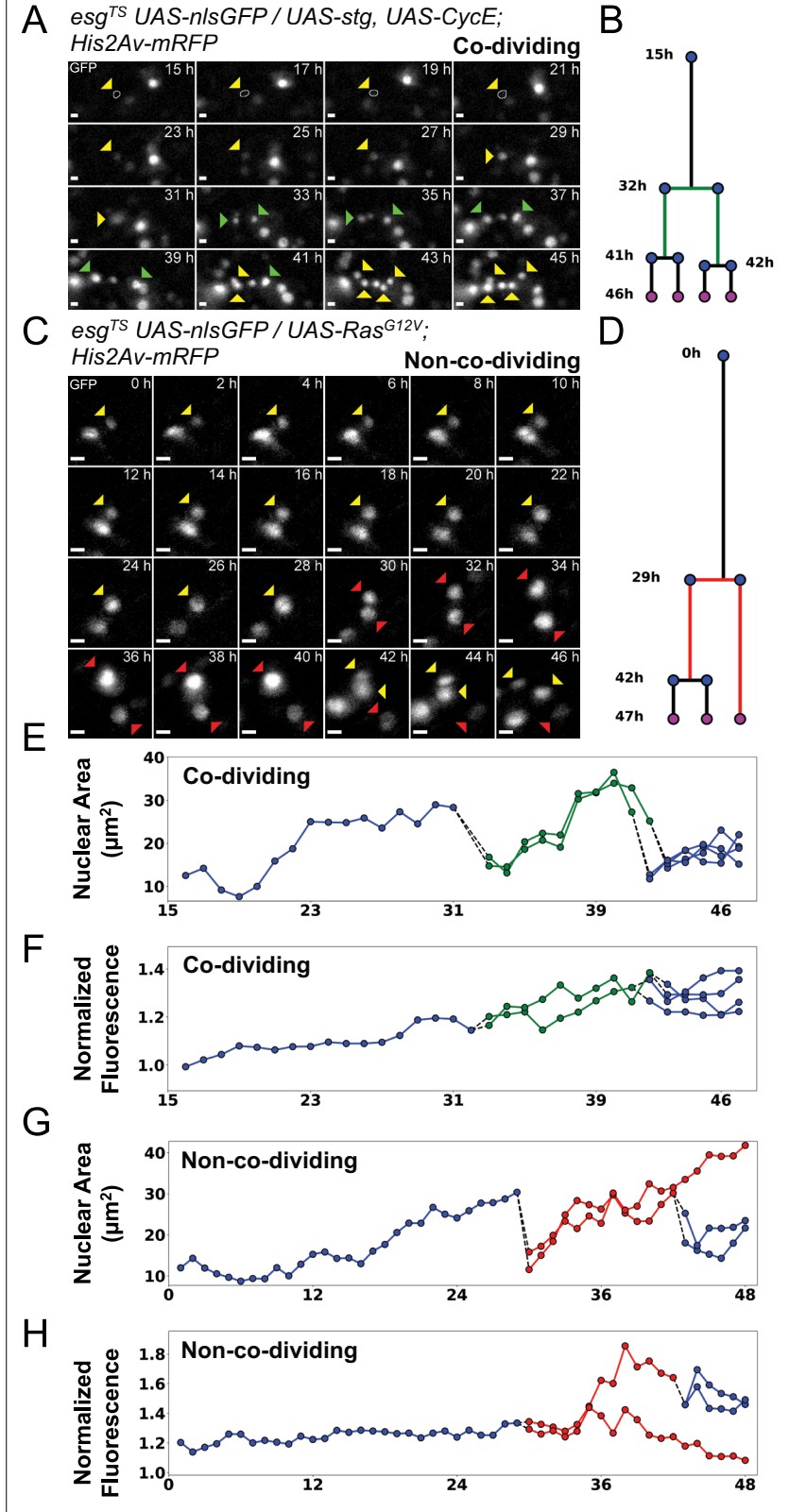

**Figure 6.** Examples of asymmetric and symmetric divisions. (**A**) nlsGFP channel showing a stem cell undergoing division upon expression of *stg* and *CycE* via *esg^TS* driver and giving rise to co-dividing siblings. The resulting daughter cells (green arrowheads) can be seen further dividing at the 41 hr and 42 hr time-points. A white circle denotes the outline of the starting cell at the initial time-points, when signal was weakest. Images are maximum

*Figure 6 continued on next page*

*Figure 6 continued*

intensity projections of z-slices encompassing the cells in the lineage. See also *Figure 6—video 1*. Scale bar is 5 µm. (**B**) Lineage diagram for cell in A. Green branch denotes co-dividing siblings. Cell present at the end of the imaging session are marked by magenta dots. (**C**) nlsGFP channel showing a stem cell undergoing division upon expression of $Ras^{G12V}$ via $esg^{TS}$ driver giving rise to non-co-dividing siblings. Of the resulting daughter cells (red arrowheads), only one can be seen dividing further. The non-dividing cell is observed for at least 6 hr following the division of its sister cell. Images are maximum intensity projections of z-slices encompassing the cells in the lineage. See also *Figure 6—video 2*. Scale bar is 5 µm. (**D**) Lineage diagram for cell in C. Red branch denotes non-co-dividing siblings. Cell present at the end of the imaging session are marked by magenta dots. (**E–H**) Plots of nuclear area (**E, G**) and nuclear GFP mean intensity (**F, H**) for the lineage in A (**E–F**) and C (**G–H**). Dotted lines connect mitotic cells to their progeny. Green and red lines denote co-dividing and non-co-dividing siblings, respectively. The GFP channel was used for quantification.

The online version of this article includes the following video and source data for figure 6:

**Source data 1.** Raw data for *Figure 6E–H*.

**Figure 6—video 1.** Example of division giving rise to co-dividing siblings.
https://elifesciences.org/articles/76010/figures#fig6video1

**Figure 6—video 2.** Example of division giving rise to non-co-dividing siblings.
https://elifesciences.org/articles/76010/figures#fig6video2

---

of progenitor cells that divided in response to *stg* and *CycE* co-expression (~25%; *Figure 4D*) was not significantly different from that observed after forced expression of $Ras^{G12V}$, suggesting that all receptive progenitors are activated in both cases. However, co-dividing siblings appeared in 6 of 12 ISC lineages that overexpressed *stg* and *CycE* (*Figure 4E*), a significantly higher frequency to the 1 of 17 observed after $Ras^{G12V}$ overexpression (p=0.0106, Fisher's exact test). We detected no significant difference in cell cycle duration in progenitors' cell cycles driven by *stg* and *CycE* (9.4±4.6 hr) and cell cycles driven by $Ras^{G12V}$ (8±2.76 hr; *Figure 4F*), suggesting that their differing abilities to produce co-dividing sibling cells may reflect different effects on the differentiation process.

## Co-dividing sibling cells actively move apart

Combining all the lineages described above, we were able to identify eight divisions that yielded co-dividing siblings (see *Figure 6A–B* and *Figure 5—video 1* for an example). For seven of these, sibling cells divided within 2 hr of each other, while for one pair the interval was 5 hr. Based on this, we looked for divisions where one sibling cell was seen dividing further, while the other remained quiescent for the remainder of the experiment, which had to last at least 6 hr after the sister's mitotic event (i.e. the second division in the lineage). For simplicity, pairs of siblings with this behavior will be henceforth be referred to as 'non-co-dividing siblings'. These are most likely asymmetric divisions that give rise to an ISC/EB pair. However, since specific cell fate markers were not assayed in our experiments, this cannot be verified. Moreover, we cannot exclude that both cells would not eventually divide, given enough time. Regardless, the behavior of these daughter pairs was indeed distinct from that of co-dividing siblings described above. Following this definition, we classified 17 sibling pairs as non-co-dividing (see *Figure 6D–F* and *Figure 6—video 2* for an example), 9 from $Ras^{G12V}$- and 8 from *stg* and *CycE*-expressing guts. Interestingly, for 8 of these 17 pairs, the non-dividing cells displayed increases in nuclear size over time, and also lost GFP expression, which is indicative of differentiation towards the EC cell fate (*Figure 6C and G–H*). This behavior was not observed for co-dividing siblings, where both cells increased their nuclear size before dividing further, but maintained GFP expression (*Figure 6A and E–F*). Hence, future studies using our culture and imaging techniques in conjunction with differentiation markers should be able to discern new details about ISC differentiation.

It was previously reported that spindle orientation is indicative of whether an ISC mitosis is asymmetric or symmetric, with symmetric divisions being planar relative to the visceral muscle layer, and asymmetric divisions displacing one daughter cell apically, away from the visceral muscle (*Hu and Jasper, 2019*; *Ohlstein and Spradling, 2007*; *Goulas et al., 2012*). As we restricted the interval between subsequent imaging frames to 1 h to reduce phototoxicity, we were not able to measure spindle orientations. However, it has been proposed that the orientation of the mitotic spindle results in daughter cells residing at different levels within the pseudostratified intestinal epithelium, with newborn EBs being more apical (*Ohlstein and Spradling, 2007*). We therefore selected mitotic events

that occurred in regions of epithelium that were flat with reference to the imaging plane (based on cell nuclei positions). The 3D profile of sister cells after mitosis was then reconstructed and we measured the angles between sister cells. Measured angles were found to be <15° in 29/33 cases for $Ras^{G12V}$- and in 30/32 cases for $stg$ +$CycE$-induced mitoses. Moreover, no significant differences in sister cell angles were observed between co-dividing and non-co-dividing siblings (*Figure 7A–C*). These observations do not support that view that apical displacement of daughter cells is associated with EB fate specification. However, as we did not image the mitotic spindle during mitosis, it is possible that spindles were oriented differently in divisions giving rise to the two sibling pair types, but that this difference in orientation was lost after cytokinesis.

A significant difference between co-dividing and non-co-dividing siblings was found, however, when we tracked newborn cells over time. Due to some cells having cell cycles lasting less than 8 hr, we only considered the first six 1 hr time-points following the appearance of a sister pair (defined as time-point 0). Measuring the distance between sister cells at each time interval, we found that cells in non-co-dividing pairs remained close to one another until the following mitotic event (*Figure 7D*, red line). Cells in co-dividing pairs, however, moved apart from one another (*Figure 7D*, green line). These behaviors could be observed even when considering unequal and co-dividing siblings in the same lineage (*Figure 7E–F*). If co-division and non-co-division are indeed indicative of symmetric and asymmetric division, respectively, this observation suggests that commitment to differentiation, as occurs after asymmetric divisions, requires that sister cells remain in contact for ~3–5 hr (*Figure 7D and F*, red line). This is in line with previous measurements of *Notch* activation dynamics in differentiating EBs (*Martin et al., 2018*). Conversely, rapid separation of sister cells following a division may be necessary to generate a symmetric division that duplicates ISCs.

Differences in the motility of co-dividing and non-co-dividing siblings could explain these observed effects. We therefore measured cell motility as the distance in 3D space that a cell travelled between one time-point and the next. We found that the genotype used to induce proliferation did not have an effect on the cell motility of either sibling type (*Figure 7—figure supplement 1A*). Likewise, no difference in cell motility was found when considering the effect of time on cell movement of non-co-dividing (*Figure 7—figure supplement 1B,C*) or co-dividing (*Figure 7—figure supplement 1D*) siblings. Therefore, we directly compared the motility of both pair types irrespective of genotype or time-point analyzed (*Figure 7—figure supplement 1E*), but also found no significant difference. This indicates that cells of both pair types migrate within the epithelium at similar speeds. Since non-co-dividing siblings tend to remain close to one another over time, their movement is most likely random. On the other hand, since the distance between co-dividing siblings increases over time, their movement is likely more directional, such that cells in a sibling pair move away from each other. It would therefore be expected that, when considering co-dividing pairs, the relative movement of one cell to its sister should be greater than that observed for non-co-dividing pairs. We therefore considered the motion of sister pairs, decomposing their movement in X- and Y-axis components and summed the resulting vectors, thus calculating the movement of a cell relative to its sister along the X- and Y-axis. As expected, the magnitudes of the reconstructed relative movements between co-dividing pairs was significantly greater than that for non-co-dividing pairs (*Figure 7—figure supplement 1F*). This confirms that the movement of co-dividing cell pairs was directional, with the two cells moving away from one another right after division.

## Ex vivo culture of other adult *Drosophila* organs

To test the feasibility of our culture setup to sustain other adult *Drosophila* organs, we first focused on the Malpighian (renal) tubules. Being physically connected to the intestine, we reasoned the two organs may share similar requirements for their survival ex vivo. A shared characteristic between Malpighian tubules and the adult midgut is the presence of a population of progenitor cells marked by *esg* expression (*Singh et al., 2007*; *Wang and Spradling, 2020*). Using the $esg^{TS}$ driver system, GFP expression could be induced in tubules cultured at 18 °C for 24 hr and then shifted to the permissive temperature (29 °C), indicating their long-term survival (*Figure 8A–B*). Moreover, we found that Malpighian tubules cultured for 3 days could still contract regularly (*Video 2*), albeit only if still attached to intestines.

A key component of our culture system that prolongs the viability of midguts is the co-culture with dissected abdomens and ovaries. On closer inspection, we found that adult hearts (the dorsal vessel),

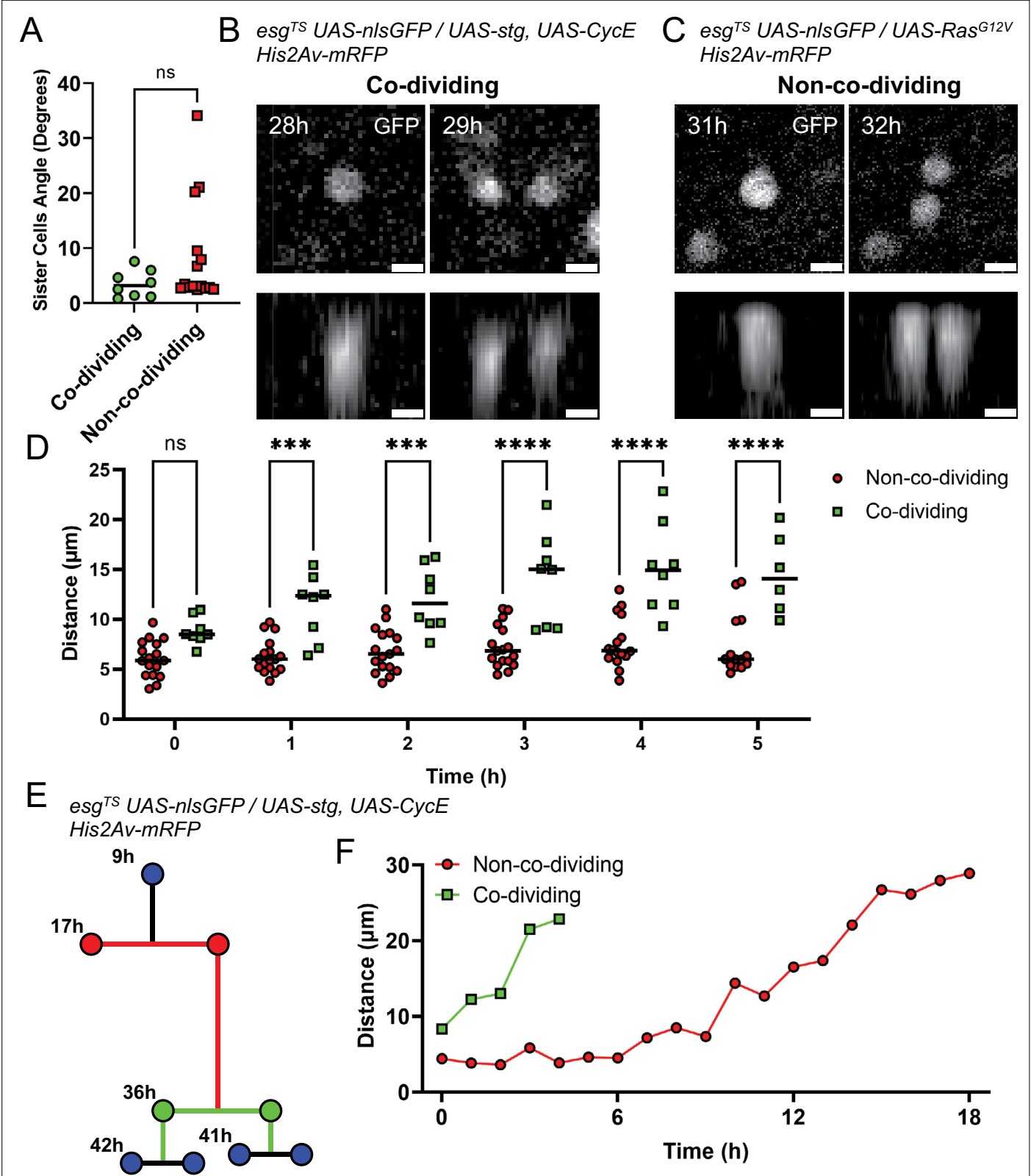

**Figure 7.** Analysis of non-co-dividing and co-dividing siblings in the *Drosophila* midgut. (**A**) Angle between daughter cells after mitosis as referenced to the imaging plane. No difference was found between the two types of daughter pairs (Mann-Whitney). (**B**) XY (top panel) and Z (bottom panels) nlsGFP profiles of the division event from **Figure 6A and B** giving rise to co-dividing siblings. Cell proliferation was stimulated by *esg^{TS}*-driven expression of *stg* and *CycE*. Scale bar is 5 μm. (**C**) XY (top panel) and Z (bottom panels) nlsGFP profiles of the division event from **Figure 6C and D** giving rise to non-co-

*Figure 7 continued on next page*

*Figure 7 continued*

dividing siblings. Cell proliferation was stimulated by *esg^TS*-driven expression of *Ras^G12V*. Scale bar is 5 µm. (**D**) Internuclear distance between daughter cells in the first 5 hr after mitosis. Time point 0 hr denotes the first at which the two daughter cells are visible. Error bars represent standard deviation (Two-way Anova and Šídák's multiple comparisons test). (**E**) Example of lineage characterized by both non-co-dividing (red) and co-dividing siblings (green). Cell proliferation was stimulated by *esg^TS*-driven expression of *stg* and *CycE*. (**F**) Internuclear distance between the non-co-dividing and co-dividing sister cells from the lineage in E. Time point 0 hr denotes the first at which the two daughter cells are visible. (ns, not significant, ***, p<0.001; ****, p<0.0001).

The online version of this article includes the following source data and figure supplement(s) for figure 7:

**Source data 1.** Raw data for *Figure 7A, D and F*.

**Figure supplement 1.** Lack of effect of genotype and time-point after mitosis on cell motility.

**Figure supplement 1—source data 1.** Raw data for *Figure 7—figure supplement 1A-F*.

which lay along the abdominal cuticle, could be seen still beating regularly after up to 10 days in culture (*Video 3*). Similarly, the muscle sheet that envelopes the ovaries still contracted after 3 days in culture (*Video 4*). We then took a closer look at ovaries by dissecting individual ovarioles for live-imaging. A distinctive characteristic of stage 1–8 follicles is their rotation within their follicle cell sheath, along their long axis (*Cetera et al., 2014*). In our ex vivo cultures, we routinely observed rotating stage 4 follicles that grew in size and started to elongate (*Figure 8C–C'* and *Figure 8—video 1*), indicative of progression to stage 5 (*Cetera et al., 2014*). Notably, this phenomenon continued for up to 48 hr, suggesting the long-term survival of follicles in our explants.

Therefore, we believe our culture protocol could be applied to other adult *Drosophila* tissues and will be useful in investigating a wide range of biologically relevant phenomena.

## Discussion

The adult *Drosophila* midgut has emerged as a powerful tool to understand the biology of epithelia and their resident stem cells. In recent years this system has been enriched by the development of advanced live-imaging approaches (*Deng et al., 2015*; *Xu et al., 2017*; *He et al., 2018*; *Martin et al., 2018*; *Hu and Jasper, 2019*) that allow the observation of adult midguts for up to 16 hr. However, many biologically interesting processes, for instance regeneration, occur over longer time-spans, and so methods for extended culture and live-imaging should prove advantageous.

Several factors could cause the limited survival of midguts ex vivo. Firstly, currently available *Drosophila* culture media are based on larval hemolymph composition. We have shown here that minimal modifications to Schneider's medium are sufficient to reduce cell death ex vivo (*Figure 2A*). Midguts may also receive nutrients and signaling molecules from other organs such as ovaries and fat body, which are in close proximity to the intestine. Indeed, using fly extract as a culture medium and co-culturing intestines with ovaries and fly abdomens resulted in dramatic decreases in cell death (*Figure 2A*). Proper oxygenation is also a concern as it had been found to be essential for other *Drosophila* organ ex vivo cultures (*Strassburger et al., 2017*). Indeed, trachea ramify throughout the fly's internal organs, and in the intestine they even reach through the visceral muscle to contact epithelial cells directly (*Li et al., 2013*). Therefore, we designed our culture setup to keep guts elevated and close to the surface of the culture medium at a liquid-air interface (*Figure 1*), which we found to be important for proper tissue oxygenation. Moreover, the sample setup was designed to be efficient and simple to construct for ease of reproducibility, allowing up to 12 explanted midguts to be imaged in parallel in a single dish. Lastly, prolonged live-imaging sessions can be hampered by phototoxicity. This can be resolved by reducing the intensity of the excitation light and exposure times, albeit at the cost of reduced signal/noise ratios and frame rates. Controlling each of these factors has allowed us to culture healthy explanted midguts for up to 3 days ex vivo (*Figure 2C* and *Figure 2—video 1*), and other organs for even longer periods.

Using our system, we observed that, while midguts maintain their ability to respond to tissue stress ex vivo, progenitor cells in undamaged intestines are quiescent. Previous estimates of mitotic rates based on immunostaining for the mitotic marker phospho-Ser 10-Histone 3 generally showed a wide range of baseline values, with numbers as low as 1–3 mitoses per midgut (*Jiang et al., 2009*). Nonetheless, even considering a low mitotic rate and using an estimate of mitosis duration (*Martin et al., 2018*) and ISCs numbers (*Ohlstein and Spradling, 2006*; *Choi et al., 2011*; *O'Brien et al., 2011*; *Jin*

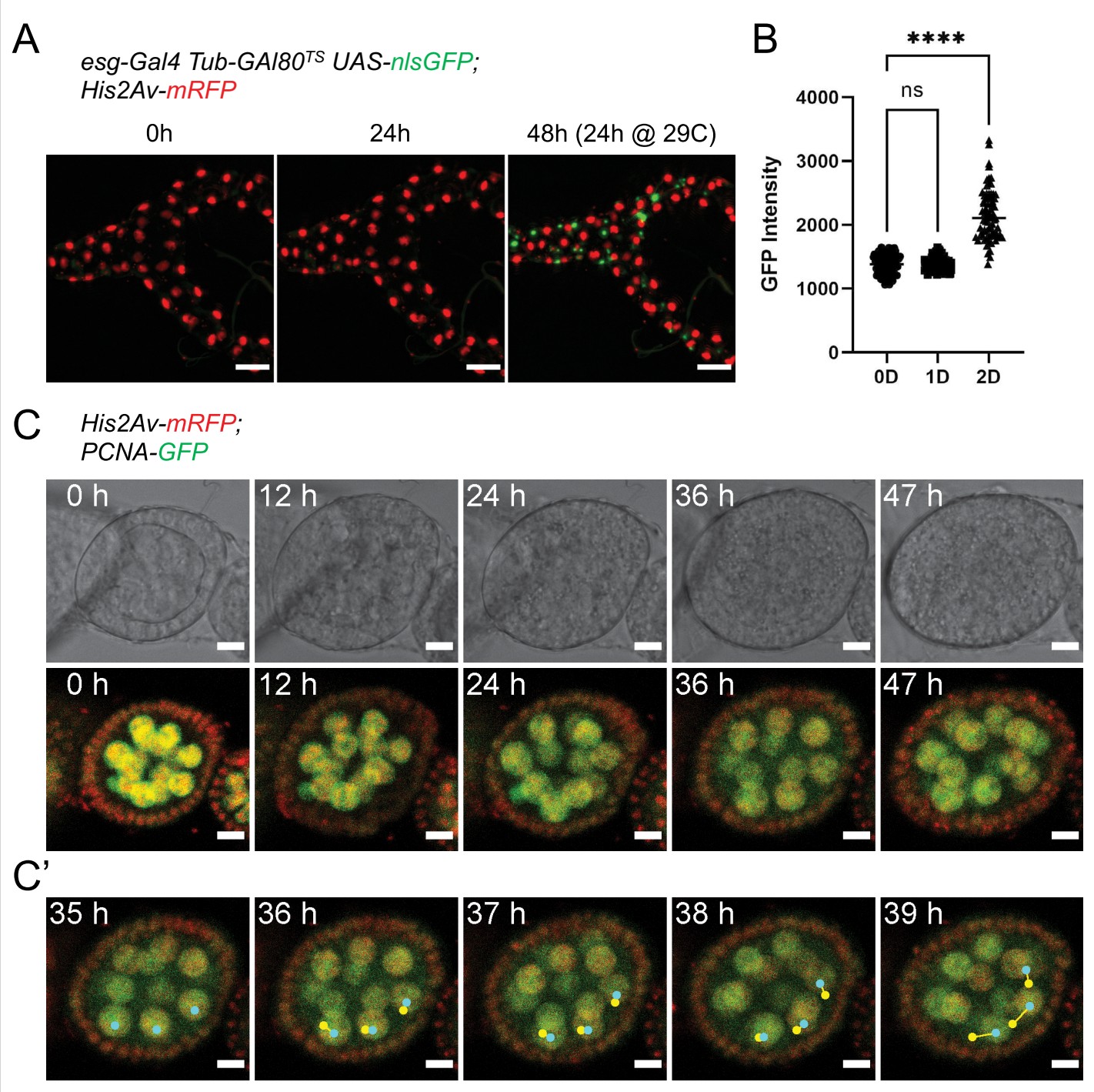

**Figure 8.** Ex vivo culture of Malpighian tubules and ovaries. (**A**) Malpighian tubules cultured at 18 °C for 24 hr, then shifted to 29 °C are still able to activate expression of nlsGFP driven by the *esg^TS* system, showing that the epithelium remains healthy long-term. Images are maximum intensity projections. Scale bar is 50 μm. (**B**) Quantification of GFP expression of progenitor cells from Malpighian tubules cultured ex vivo. Same cells were measured at 0, 1, and 2 days after explantation. (Dunnett's multiple comparisons test) (ns, not significant, ****, p<0.0001) (**C**) Stage 4 follicle growing in size and elongating over the course of 2 days ex vivo. (**C'**) Selected frames showing follicle rotation. Cyan dots mark the current position of a nucleus, while yellow ones marked the position in the previous frame. Images are maximum intensity projections of the 4 center-most z-slices. See also *Figure 8—video 1*. Scale bar is 10 μm.

The online version of this article includes the following video and source data for figure 8:

**Source data 1.** Raw data for *Figure 8B*.

*Figure 8 continued on next page*

*Figure 8 continued*

**Figure 8—video 1.** Example of a rotating stage 4 follicle progressing to stage 5 ex vivo and starting to elongate along its long axis.

https://elifesciences.org/articles/76010/figures#fig8video1

*et al., 2017*), we expected to see several mitoses even in undamaged intestines (e.g. >5 for fields with 50 or more *esg⁺* cells). As ISC proliferation could be stimulated by tissue damage ex vivo, this suggests that, in homeostatic conditions, stem cells may only proliferate when the need to replace damaged or dying cells arises. Given the lack of cell death in undamaged midguts ex vivo, the previously reported proliferation-suppressive effect of enterocytes (*Liang et al., 2017*; *Jin et al., 2017*) may be responsible for the lack of observed mitotic events. Interestingly, we also did not observed differentiation events in undamaged intestines, which suggest that enteroblasts are a stable cell type, rather than being transient progenitors that are present only during periods of rapid ISC division, and are rapidly lost via differentiation or apoptosis (*Reiff et al., 2019*). Indeed, a previous analysis of the EB gene expression profile showed the existence of EB-specific genes, consistent with EBs being a distinct cell type (*Dutta et al., 2015*; *Hung, 2020*). As a consequence, the enteroblast pool may constitute a first line of response to tissue damage, that rapidly differentiate to generate new ECs, while buying time for stem cells to progress through the cell cycle.

Our culture system can also be used in combination with temperature-sensitive gene induction or knock-down tools, thus expanding its applications. When genetically stimulated by the expression of constitutively active $Ras^{G12V}$, ISCs proliferated rapidly (*Figure 4* and *Figure 4—video 2*). Similarly, co-overexpression of *stg* and *CycE* also resulted in ISC proliferation. These are strong genetic manipulations known to promote ISC proliferation. Glycine 12 mutations in KRAS, which result in the constitutive activation of the small GTPase, are among the most frequent mutations in colorectal and other cancers (*Prior et al., 2020*). In the adult *Drosophila* intestine, this same mutation (G12V) has been shown to drive ISC proliferation (*Jiang et al., 2011*; *Jin et al., 2015*), and EB growth and endoreplication (*Xiang et al., 2017*). Moreover, a recent study suggested that a small subset of EBs could also be induced to proliferate by $Ras^{G12V}$ (*Tian et al., 2021*). Co-overexpression of *stg* and *CycE* can also promote ISC proliferation by directly stimulating cell cycle progression (*Kohlmaier et al., 2015*). Similarly to $Ras^{G12V}$, this genetic stimulation could also drive

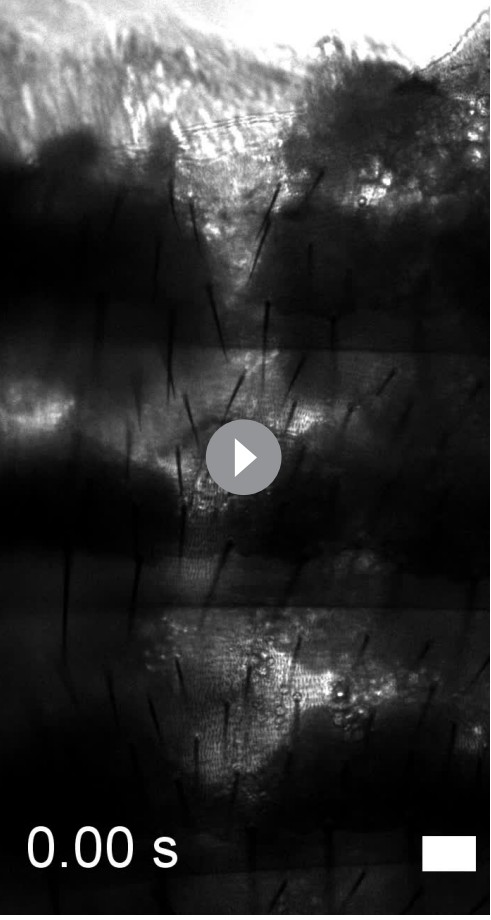

**Video 3.** Example of heart lining the dorsal side of a dissected fly abdomen beating after 10 days ex vivo. Scale bar is 50µm.

https://elifesciences.org/articles/76010/figures#video3

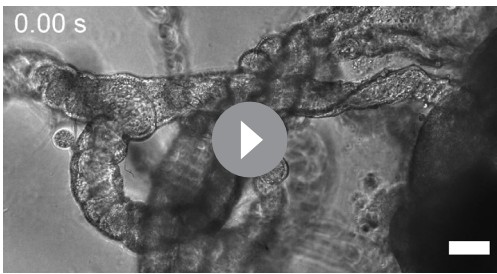

**Video 2.** Example of Malpighian tubule contracting after 3 days ex vivo. Note that the tubule is still connected to the midgut. To avoid inhibiting contractions, isradipine was not supplemented, resulting in the midgut rupturing near the imaged site, releasing visible debris. Scale bar is 50µm.

https://elifesciences.org/articles/76010/figures#video2

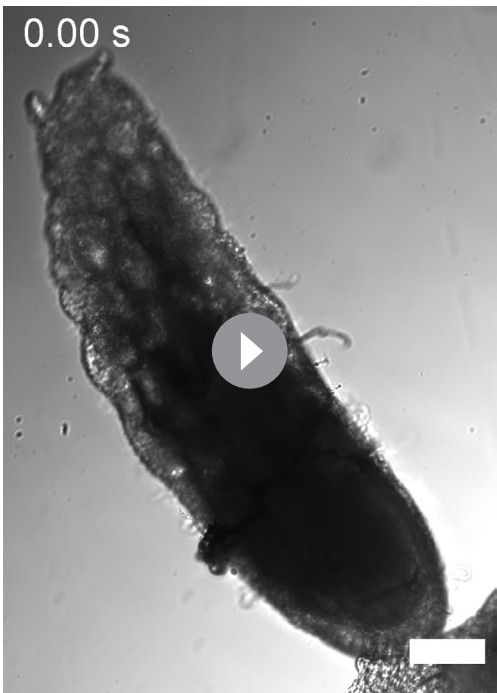

**Video 4.** Example of ovary contracting after 3 days ex vivo. Scale bar is 100μm.

https://elifesciences.org/articles/76010/figures#video4

EB proliferation, albeit not as strongly as in ISC. It's reasonable to assume that the EBs responsive to either *Ras^G12V* or *stg* and *CycE* co-expression may be immature and still close to a stem cell state. When reconstructing cell lineages induced by these genetic manipulations, we observed divisions that gave rise to daughter cells with two distinct behaviors: (1) 'co-dividing' pairs in which both daughter cells divided again, like the progeny of symmetric divisions; and (2) 'non-co-dividing' pairs in which only one daughter cell divided while the other did not for the remainder of the imaging session (>6 h), as would be the case for the progeny of asymmetric divisions. By analyzing the reconstructed lineages, we found that upon *Ras^G12V* stimulation most lineages did not present co-dividing siblings. Interestingly, if co-dividing siblings were the result of symmetric divisions, this would be in accordance to the previously described prevalence of asymmetric division events in normal intestines (*de Navascués et al., 2012*; *O'Brien et al., 2011*). This could suggest that the EGFR-Ras-Erk pathway may have a role in differentiation. Indeed, several progenitor cells, when stimulated by *Ras^G12V*, did not divide, but rapidly grew in nuclear size and lost *esg* expression, which is indicative of EB to EC differentiation and a similar phenotype to what previously reported (*Xiang et al., 2017*). Alternatively, Stg and CycE overexpression may suppress differentiation, resulting in the increased number of symmetric-like divisions we observed (*Figure 4E*).

One major difference that we did observe between co-dividing and non-co-dividing siblings was in the behavior of sister cells. Non-co-dividing siblings remained close to one another for several hours after mitosis. This is significant as it is known that cell-cell contacts between progenitor cells are required for promoting differentiation (*Ohlstein and Spradling, 2006*; *Micchelli and Perrimon, 2006*). Indeed, interactions between Notch on the surface of the EB and Delta expressed on the ISC surface is a strong promoter of EB to EC differentiation (*Ohlstein and Spradling, 2006*; *Micchelli and Perrimon, 2006*; *Ohlstein and Spradling, 2007*). It was previously shown that EB differentiation via N activation requires several hours to resolve (*Martin et al., 2018*). This time frame matches our observations, which show non-co-dividing sister pairs remaining in contact for at least ~3–5 hr after division. Adherens junctions may be affecting these cell contacts as strong levels of shotgun (e-cadherin) and armadillo (β-catenin) are found in between ISC and EB pairs (*Choi et al., 2011*). As EGFR signaling is known to impact adherens junctions remodeling (*Buchon et al., 2010*; *O'Keefe et al., 2007*; *Robertson et al., 2012*), this could help explain the prevalence of non-co-dividing siblings in cell lineages stimulated by *Ras^G12V*

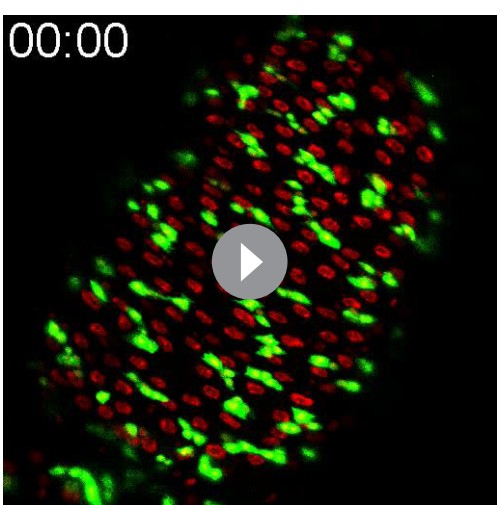

**Video 5.** Example of high frame-rate long-term confocal imaging. Undamaged control intestine expressing His2Av.mRFP (red) and esg^TS-driven nlsGFP (Green) induced 24h prior to imaging. Maximum intensity projection. Scale bar is 50μm.

https://elifesciences.org/articles/76010/figures#video5

expression. Co-dividing pairs had the opposite behavior, and moved apart right after division. If co-dividing siblings were the result of symmetric division expanding the ISC pool, this would result in the dispersion of stem cells through the midgut. As symmetric divisions are known to occur during adaptive growth of the intestine, especially in the days after eclosion, this behavior could help explain how ISCs space themselves uniformly in the epithelium (*O'Brien et al., 2011*; *Ahmed et al., 2020*).

Despite these successful applications, the system we developed still has limitations. The optimized dissection procedure limits midgut damage, but does not completely eliminate the risk. The agarose pads, albeit thin, can interfere with high powered objectives with short working distances (e.g. 40 X and above). Using thinner agarose pads could help to reduce the required working distance, but may result in hypoxia. This in turn could be solved by increasing the oxygen concentration using microscope incubation chambers equipped with an atmosphere control unit. Midgut survival ex vivo also seems to be limited by the growth of enteric bacteria, especially since, once dissected, midguts cannot properly move food through the intestine and defecate. Indeed, we observed that the visceral muscle, which is not in direct contact with luminal contents, could survive and contract regularly for up to a week in optimal conditions, despite the death of the adjacent epithelium. Generating axenic flies may help to further extend the survival of the midgut epithelium ex vivo. Newer, gentler imaging technologies such as light-sheet microscopy could also improve survival during live-imaging sessions by reducing phototoxicity. It s also possible that different organs may have specific requirements in terms of media composition and additives, in which case tailoring culture media to a specific organ may be beneficial. Indeed, even the same organ can have different requirements in male and female flies (*Ahmed et al., 2020*; *Hudry et al., 2016*; *Hudry et al., 2019*). Lastly, to limit mechanical damage to the intestines and to more easily image them, we used the calcium blocker isradipine to inhibit visceral muscle peristalsis. The lack of peristaltic contraction may have negative effect on midgut biology, aside from the retention of food and proliferating bacteria in the lumen. Indeed, it was previously reported that ISCs are sensitive to mechanical stimuli (*He et al., 2018*). If food passage itself can stimulate ISC proliferation, its inhibition could partially help to explain the lack of mitosis in healthy midguts ex vivo.

Nonetheless, the increased survival ex vivo our protocol allows is significant and we believe it can enable experiments that will lead to a better understanding of the mechanisms that mediate epithelial homeostasis, such as the regulation of asymmetric and symmetric ISC division events. Our protocol may also provide a platform to dissect inter-organ interactions, given the positive effect that co-culture with ovaries and fat bodies had on midgut survival (*Figure 2A*). Moreover, Malpighian tubules, hearts, and ovaries did show increased survival when cultured with our protocol (*Figure 8* and *Figure 6—video 1*, *Figure 6—video 2*, *Video 2*, *Video 3*). Finally, we believe that the possibility to visualize the effects of gene induction or silencing in real-time using fluorescent markers will be very useful for dissecting the roles of specific signaling pathway components and in modeling human disease.

## Materials and methods
### *Drosophila* stocks

$w^{1118}$ (Bloomington *Drosophila* Stock Center 3605)
*esg-Gal4 tubGal80ts UAS-GFP / CyO; UAS-flp Act >CD2>Gal4/TM6 B* (PMID: 19563763)
esg-Gal4 Tub-Gal80$^{TS}$ UAS-nlsGFP / CyO
mex-Gal4 Tub-Gal80$^{TS}$ UAS-GFP / CyO
esg-Gal4 UAS-GFP
esg-Gal4 Tub-Gal80$^{TS}$ UAS-nlsGFP / CyO; His2Av-mRFP / TM6B
*His2Av-mRFP* (Bloomington *Drosophila* Stock Center 23650)
*UAS-N$^{RNAi}$ / TM6B* (PMID: 26237646)
*UAS-Ras$^{G12V}$ / CyO* (PMID: 21167805)
*UAS-stg, UAS-CycE* (PMID: 24975577)
*PCNA-GFP* (Stefano Di Talia, Duke University Medical Center, USA)
His2Av-mRFP; PCNA-GFP

## Fly rearing

Flies were raised on standard cornmeal and molasses fly food. Prior to dissection, flies were flipped to fresh vials without live yeast daily for 3 days at 18 °C to reduce the load of commensal bacteria. On the morning of the dissection, flies were fed a sucrose 0.05% aqueous solution on a cotton pad at room temperature (25 °C) for 4–6 hr to clear most luminal contents. This also helped to reduce accumulation of food in the posterior section of the gut, which could lead to the mechanical stress of the epithelium. For SDS feeding experiments, flies were fed overnight either standard food or food mixed with SDS to a final concentration of 0.2% v/w. Both foods were also mixed with a blue food-safe dye to control for feeding. The following morning, flies were fed a sucrose 0.05% aqueous solution on a cotton pad to clear most of the luminal SDS.

## Modified Schneider's medium for adult *Drosophila* tissues

Schneider's medium (Genesee Scientific, 25–515) was modified by adding the following reagents to the stated final concentrations: 1 mM trisodium citrate dihydrate (ThermoFisher Scientific, BP327), 91.2 mM sodium chloride (Sigma-Aldrich, S9888), 55.8 mM D-trehalose anhydrous (Sigma-Aldrich, T0167), 10 mM glutamine (Gibco, 25030), and 2 mM N-acetyl cysteine (Sigma-Aldrich, A7250) (*Table 1*). Glutamine needs to be added only if the batch of Schneider's medium used is glutamine-free. Medium was then filtered using 0.22 µm syringe filters (VWR, 28145) and stored at 4 °C. See *Supplementary file 1* for recipe. This medium was used without additives during dissection, agarose gel stock preparation, and as a base for fly extract.

For fly extract, well-fed female flies were anesthetized on ice. Using mortar and pestle, flies were homogenized on ice in the presence of 10µ$\lambda$ per mg of flies of modified Schneider's medium (as described above) with bovine serum albumin (BSA) added to 1%. The homogenate was centrifuged at 0.6 G and 4 °C for 10'. Supernatant was saved and fly carcasses discarded. The centrifugation step was repeated 3 times until all solid fly residues were eliminated. Extract was heat inactivated by heating at 60 °C for 5', then centrifuged at 0.6 G and 4 °C for 10'. Supernatant was saved and filtered using 0.22 µm syringe filters. Extract was aliquoted and stored at –20 °C before use.

For live-imaging, 100% fly extract prepared in modified Schneider's medium was used as a base for the complete culture medium. Fly extract was slowly thawed at 4 °C, then 10% fetal calf serum (Gibco, 26140079), 1:100 Antibiotic-Antimycotic (ThermoFisher Scientific, 152400062), 100 µg/ml Ampicillin (Fisher Scientific, AC611770250), and 25 µg/ml Chloramphenicol (Fischer Scientific, BP904-100) were added. To suppress peristaltic movements, 10 µg/ml isradipine (SigmaAldrich, I6658) was added to the complete medium immediately before imaging.

## Sample preparation for long-term culture and live imaging

1. Prior to dissection, a 35 mm dish with lockable lid (ibidi, µ-Dish 35 mm low, 80136) is prepared by first placing a thin wet paper tissue around its inner rim to reduce evaporation during long imaging sessions (*Figure 1O*, left panel);
2. Agarose pads are then cast by spreading 2 µl of low gelling temperature agarose (Sigma Aldrich, A9414), heated to 70 °C, over four 4 mm areas in the observation region of the dish. For each dish, 4 pads can be easily cast (*Figure 1O*, left panel). The agarose solution is prepared from powder as a 1% stock in modified Schneider's medium without additives and stored at 4 °C in 200 µl aliquots. Aliquots can be melted and re-gelled several times, provided evaporation is not excessive;
3. Dishes are then stored at room temperature while midguts are isolated by dissection;
4. A small amount of medium is then added to the top of each agarose pad to facilitate the transfer of midguts. These are transferred very carefully, by holding them in a drop of liquid in between the grasping ends of a forceps. The drop is then touched to the top of an agarose pad, gently depositing the midgut trapped in it. Care must be exercised to ensure that the midgut rests entirely within the drop of liquid, without touching the dry surfaces of the forceps, to which it may stick. Each agarose pad can house up to 3 guts;
5. Once all midguts have been transferred, liquid from the top of the agarose pads is removed as much as possible using forceps, while leaving a small amount to avoid desiccation of the intestines. Midguts are then gently repositioned for proper imaging, if required;

6. Guts are then covered with a thin layer of low gelling temperature agarose 0.5% cooled to 37 °C (*Figure 1O*, middle panel). The layer must be just enough to cover the midguts' surface (~1 µl per pad);

7. The sample is incubated for 5 min at room temperature before the agarose structures are connected between them and to the sides of the observation area by creating agarose bridges with 0.5% low gelling agarose (*Figure 1O*, middle panel). This increases the stability of the overall sample, facilitating its transport in case the sample has to be prepared at some distance from the microscope that will be used to image it. If required, agarose domes can also be strengthened with an additional thin layer of agarose;

8. After 10 min, the agarose will have solidified and 120 µl of complete culture medium can be carefully added to the sample (*Figure 1O*, right panel). The small volume is enough so that all midguts will receive nutrients throughout the culture duration, while ensuring that the upper-most surface of the agarose structures is not submerged by liquid, creating a liquid-air interface;

9. Finally, ovaries and fly abdomens that were dissected along with the midguts are added to the culture, free-floating in between the agarose pads. The sample will thereby be ready for imaging;

## Optimized dissection to avoid damaging of midguts

1. To reduce the risk of contamination of the culture by bacteria residing on the animal exterior, $CO_2$ anesthetized flies are surface sterilized by submerging them in 70% ethanol for 2 min and then in 50% bleach for 1 min. Most flies survive this treatment;

2. Flies are then washed and stored in 1 X PBS modified as they are dissected one by one in modified Schneider's medium without additives. Since this step is fast (<2 min), using complete medium based on fly extract is not required;

3. Using micro-scissors the head is removed with a clean cut, thus ensuring that the crop and proventriculus still reside in the fly thorax (*Figure 1B and C*; *Figure 1—video 1*);

4. The cuticle around the anus is pulled, exposing the hindgut (*Figure 1D*);

5. Holding the fly gently with forceps around the thorax-abdomen junction, the soft ventral abdominal cuticle is ripped using another forceps, pulling it along the length of the fly towards the anus, thus exposing the midgut (*Figure 1E and F*; *Figure 1—video 1*);

6. The abdomen is then gently separated from the thorax (*Figure 1G*; *Figure 1—video 1*);

7. The crop is gently pinched and pulled out of the thorax, thus freeing the anterior midgut along with it (*Figure 1H and K* yellow arrowhead; *Figure 1—video 1*);

8. The midgut is gently freed from the abdominal cuticle (*Figure 1J*; *Figure 1—video 1*). Care has to be exercised at this step as many trachea filaments connect the midgut to ovaries and abdominal walls. Freeing the midgut from ovaries and the abdominal cuticle is important for ease of handling and imaging and for proper oxygenation. Indeed, if these structures are kept attached to the intestine, transferring the explanted organ to agarose pads is harder and carries the risk of the midgut being covered by ovaries and the cuticle, thus limiting imaging access and creating a barrier between the midgut and the air-liquid interface;

9. Crop and malpighian tubules are cut away using micro-scissors, and the hindgut is similarly removed just below its connection to the midgut (*Figure 1K–M*; *Figure 1—video 1*). This step is necessary as both Malpighian tubules and the ampulla connected to the hindgut are quite sticky and make transferring the guts to the imaging dish difficult. If desired, however, Malpighian tubules can be transferred to agarose pads for imaging, either detached or still connected to midguts;

10. Once dissected, a midgut can then be transferred to a well containing modified Schneider's medium with 10 µg/ml isradipine. Ovaries and the fly abdomen are transferred to this well too;

11. Once all midguts have been dissected, they can be carefully transferred to the agarose pads for sample preparation. To avoid damage during transfer, it is recommended that midguts be moved by keeping them in a small drop of liquid in between the prongs of a forceps. Ovaries and fly abdomens to be co-cultured with the intestines are transferred to the space between the agarose pads, free-floating in the culture medium;

12. For localized damage experiments, midguts were poked in their posterior section using an electrolytically sharpened tungsten needle once transferred to the agarose pads. To keep guts still during the procedure, the hidgut section can be clamped with forceps held in the non-dominant hand. For this, a longer hidgut section can be preserved (see point 9). It is imperative to avoid perforating the peritrophic matrix, thus preventing the contamination of

the culture from commensal bacteria. For this, guts are not perforated perpendicularly to the epithelium, but at a~45° angle. Alternatively, guts can be carefully scratched repeatedly in the same spot to produce a tear;

## Long-term live-imaging of explanted adult *Drosophila* midguts

For imaging of midgut, hearts, and Malpighian tubules we used an inverted widefield Nikon Ti Eclipse microscope equipped with an incubation chamber (Okolab) for moisture and temperature control, a CoolSnap HQ2 camera (Photometrics), a SOLA LED light engine (Lumencore), and a Prior motorized stage. To limit phototoxicity, albeit at the cost of reduced signal/noise ratio, we used a 4 x neutral density filter, limited light intensity to 5%, and limited exposure times to 100–150ms and 200–300ms for GFP and RFP signals, respectively. Follicles were imaged using a Leica SP8 confocal microscope equipped with a white laser light source and an incubation chamber (Okolab). To limit dwell time, samples were imaged at a frequency of 700 Hz. Intestines could also be successfully imaged with the same setup. For both widefield and confocal setups, videos were captured at room temperature (25 °C) or at 29 °C for temperature-sensitive gene induction experiments using a 20 X air objective (APO 20 X NA 0.75 WD 1). Multipoint acquisition was used to image the R4-5 posterior midgut section of 12 intestines during each imaging section. For each midgut, a 45 µm Z-stack was captured using a 3 µm Z-step. Focus was manually checked and corrected during the course of the imaging session with the Nikon Ti Eclipse, or using a contrast-based autofocus routine for the Leica SP8. Frame rate was typically one full multi-channel Z-stack/midgut/hour. For the imaging of Malpighian tubules and ovaries contractions and beating hearts, the frame rate was 1 slice every 0.08ms instead.

## NucGreen analysis

For cell survival experiments, guts from $W^{1118}$ flies were cultured in media containing NucGreen (ThermoFisher Scientific, R37109) diluted 1:20. Midguts were imaged once a day for 3 days, using a 10 X air objective (Plan Fluor 10 X NA 0.3 WD 16) to capture a stitched z-stack of the whole organ with z-steps of 20 µm. Images were normalized to their median pixel value to account for changes in background, then midgut areas were manually selected from maximum intensity projections. The NucGreen signal was computed as the sum of pixel values in the selected areas, normalized to the first time-point in the series.

## GFP level quantification

Midguts were explanted from 5 to 15 days old flies reared at 18 °C and cultured at 29 °C for 24 hr. Midguts were then carefully removed from the culture setup and fixed in para-formaldehyde 6% in PBS for 30 min at room temperature. Similarly, in vivo control flies were shifted to 29 °C for 24 hr, then dissected and their midguts fixed. Fixed intestines were stained for DAPI 0.1 mg/ml (Sigma-Aldrich, 10236276001) in PBS with 0.1% Triton-X100, then mounted with VECTASHIELD antifade mounting medium (Vector Laboratories, H-1000–10). The posterior region of the intestines was imaged and the GFP and DAPI signals were thresholded using Otsu's method (*Otsu, 1979*). Regions of overlap between the two tresholded channels were used as a mask to calculate the mean GFP expression of each cell of interest in the imaged field. For each intestine, the values of each measured cell were averaged to express a mean fluorescence for the whole intestine.

For GFP induction tests, malpighian tubules were detached from midguts using micro-scissors and cultured in sandwiched agarose structures as described above. Tubules were cultured at 18 °C for 24 hr then at 29 °C. Images were captured daily. Individual GFP$^+$ cells from 48 hr images were identified in previous time-points and their GFP intensity measured.

## Cell tracking and analysis

Image analysis was performed using either ImageJ or Python (v3.7.10). Using a custom Python script (*Source code 1*), each time-lapse movie was divided at the first time point into overlapping regions of interest that were then used for automatic local registration. This helps account for local movements and deformations of the intestinal epithelium which complicate the following image analysis step. Registration was based on cross-correlation of a region of interest with the next frame of the time-series. Individual ISC and their progeny were then manually tracked using a custom ImageJ macro (*Source code 2*), selecting each cell by drawing their outline at their most in-focus z slice. By

identifying all cells between successive time-points, we avoided lineage assignment errors even with long time-intervals between frames.

Cells were then analyzed as follows:

- Lineages were reconstructed with a custom Python script (*Source code 3*) using the cell positions in the imaged field across time annotated during manual tracking. When the former approach failed due to too complex rearrangements of cells within a lineage, lineages were manually annotated instead;
- Nuclear size was measured during manual tracking from the most in-focus z-slice using an upscaled image so as to achieve sub-pixel precision;
- Mean nuclear GFP intensity was measured during manual tracking and adjusted to the median value in a 50 μm window around the cell of interest to account for changes in local background;
- Cell cycle duration was measured as the number of frames (each representing a 1 h interval) from the appearance of a cell resulting from a mitotic event and its subsequent division;
- Cell motility was defined as the distance between the positions of a cell in two subsequent frames. We used the center nuclei to mark cells' positions. Since $esg^+$ + have a high nuclear/cytoplasm ration, measuring internuclear distance is a good approximation for quantifying cell motility. A few errors during image registration resulted in improper measurements, which were then discarded. For this, outlier values were first identified using the interquartile range method, then confirmed on the registered movie file;
- For analysis of the angle between sister cells, the z-profile along the axis connecting the center of the two cells was reconstructed Python. Only cells in flat sections of epithelium in reference to the imaging plane (based on enterocytes' nuclei positions) were considered. The pixel values of the stack along a line connecting the two cells were used to reconstruct the 3D profile. Once all profiles were reconstructed, the angle between the 3D centers of the sister cells was manually computed in reference to the imaging plane. Measurements were repeated three times and averaged.

## Acknowledgements

This work was supported by the Huntsman Cancer Foundation, and National Institutes of Health Grants R01 GM124434 and R35 140900 (to BAE) and P30 CA042014. We thank the Bloomington *Drosophila* Stock Center for fly stocks. We thank the University of Utah Cell Imaging Core for its instruments and assistance.

## Additional information

### Funding

| Funder | Grant reference number | Author |
| --- | --- | --- |
| National Institutes of Health | R35 140900 | Bruce A Edgar |
| National Institutes of Health | R01 GM124434 | Bruce A Edgar |
| National Institutes of Health | P30 CA042014 | Bruce A Edgar |

The funders had no role in study design, data collection and interpretation, or the decision to submit the work for publication.

### Author contributions

Marco Marchetti, Conceptualization, Resources, Data curation, Software, Validation, Investigation, Visualization, Methodology, Writing - original draft, Writing - review and editing; Chenge Zhang, Conceptualization, Resources, Validation, Investigation, Visualization, Writing - original draft, Writing - review and editing; Bruce A Edgar, Conceptualization, Supervision, Funding acquisition, Validation, Visualization, Writing - original draft, Project administration, Writing - review and editing

**Author ORCIDs**
Marco Marchetti http://orcid.org/0000-0003-0776-0486
Bruce A Edgar http://orcid.org/0000-0002-3383-2044

**Decision letter and Author response**
Decision letter https://doi.org/10.7554/eLife.76010.sa1
Author response https://doi.org/10.7554/eLife.76010.sa2

## Additional files

### Supplementary files
• Supplementary file 1. Modified Schneider's medium and stock solutions recipes.

• Supplementary file 2. Nuclear area and nlsGFP measurements for lineages from $Ras^{G12V}$ or $stg$ and $CycE$ over-expressing midguts.

• Transparent reporting form

• Source code 1. This Python script is used for local registration of adult Drosophlila midgut fluorescent imaging time-lapses. Movies are converted to 8bit, normalized by subtracting mean and dividing by the standard deviation, converted to maximum intensity or focused projections, then divided in partially overlapping regions of interest (ROIs). Each ROI is then XY-registered via cross-correlation between frames and then the most in-focus Z slice is found. Finally, ROIs are exported.

• Source code 2. ImageJ Macro designed for the manual tracking of cells in two-channel time-lapse images of adult *Drosophila* midguts. Images must be two-channel time-lapse z-stacks, with channels 1 and 2 being His2Av.mRFP and esg$^{TS}$ > nlsGFP, respectively.

• Source code 3. This Python script is used to parse files generated by the '*Source code 2*' ImageJ macro. First, the lineage is reconstructed based on relative cell distances between frames. For each cell in frame f, a new position in frame f+1 is assigned, based on the pool of annotated cell positions in f+1. All possible permutations between f and f+1 coordinates are computed and the total distance between cell positions in frame f and their newly assigned positions in f+1 is calculated. The permutation for which this distance is minimum is then kept. If lineages are too complex (e.g. cells are moving and rearranging themselves), then the user can add two additional field to the input ".tsv" file ("CellID" and "MotherID" columns) indicating each cells ID (for each cell position) and its mother cell ID (can be stated once). The lineage data is then parsed and formatted in an easy to read format, then exported to a ".tsv" file.

### Data availability
Source Data files have been provided for Figures 2, 3, 3 - figure supplement 1, 4, 6, 7, 7 - figure supplement 1, 8.

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
