## [Editor Report]

Marchetti and colleagues present a promising, ex vivo culture method for the adult *Drosophila* midgut and other abdominal organs. Highlights include demonstrated organ viability for up to 72 hours and protocols for ex vivo injury and genetic manipulation. These advances enable the first real-time lineages tracking multiple stem cell divisions, which reveal intriguing spatial-temporal behavior patterns. The manuscript provides a thorough and thoughtful evaluation of these dynamic data and includes additional information about imaging parameters that will be invaluable to those seeking to replicate this method.

---

## [Decision Letter]

**Decision letter after peer review:**

Thank you for submitting your article "An improved organ explant culture method reveals stem cell lineage dynamics in the adult *Drosophila* intestine" for consideration by *eLife*. Your article has been reviewed by 3 peer reviewers, including Lucy O'Brien as the Reviewing Editor and Reviewer #1, and the evaluation has been overseen by K Vijay Raghavan as the Senior Editor. The following individual involved in review of your submission has agreed to reveal their identity: Karen Bellec (Reviewer #2).

Essential revisions:

1) Either substantiate or revise interpretations of symmetric/asymmetric fate outcomes.

Because the movies use guts without markers for terminal fates, distinguishing asymmetric and symmetric outcomes is particularly challenging. Reviewers 1 and 3 felt that the criteria used to classify fate outcomes as symmetric or asymmetric were not sufficient to justify exclusion of the other outcome. Can the authors justify their interpretations, for instance by using midguts that express enteroblast- and enteroendocrine-specific markers? Alternately, if supporting evidence would not be available in a reasonable time frame, the authors are instead asked to describe the data in a manner that does not assign fate outcomes to division events that are ambiguous.

2) Clarify the use of nuclear size as a measure of enteroblast differentiation.

Reviewers 1 and 2 had questions about how nuclear size was used as a measure of enteroblast differentiation. For Figure 3 and Figure 4 movies, can the authors split the esg>GFP and his::RFP channels so that the nuclei are visible? This is critical to see nuclear size. For Figures 5-7 movies, only esg>GFP(nls?) is shown; can the his::RFP channel also be included? This is critical to evaluate that loss of GFP signal is not due to the trivial reason that the cell moved out of view. Can the authors provide data about the frequency of cell differentiation in the SDS movies? Can they provide additional examples of nuclear measurements for Figure 6E-H and Figure 7D-E (only one example of each type is shown)?

3) Provide additional information about organ viability in general and about ex vivo vs in vivo damage response.

Reviewers 1 and 2 raised some questions about organ viability. Can the authors provide data for the typical viability of midguts subjected to confocal imaging at frame rates more frequent than 1 hour? Can they comment on how trachea and muscle may be altered by ex vivo culture conditions? The authors develop an ingenious SDS pulse-chase injury protocol; can the authors compare the ex vivo damage response shown in Figure 4 to an in vivo response to the same pulse-chase treatment?

*Reviewer #1 (Recommendations for the authors):*

– The terms motility and cell contact are used to discuss the data in Figure 7. However, since non-esg+ cells in the tissue are not visible, it is unclear whether the analyzed cells are autonomously motile or whether they are passively moving with or displaced by other cells. Similarly, since only cell nuclei are visible, cell-cell contact cannot be ascertained (e.g. lines 417-418). Can the authors replace "motility" and "cell contact" with terms that encompass these ambiguities, or else provide information that substantiates these interpretations?

– The text states that the dissection technique used by the authors (which is nicely illustrated in Video 1) is an improvement over the standard technique. Can the authors revise the text to clarify this comparison, for instance by describing what the standard technique is?

– In Figures 4-7, I'm confused why GFP fluorescence appears to localize to cell nuclei, not the cytoplasm, given that the genotype in the figure caption does not indicate a GFP:nls. Why is GFP used to measure nuclear size rather than the (presumably more accurate) His::RFP? Why is the RFP channel not shown?

– Which asymmetric divisions were analyzed in Figure 7 (all 25 asymmetric divisions, or a subset)? Can the authors revise Figure 7D to show individual data points rather than averages?

– Peristaltic contractions: Do guts remain viable if isradipine is not included in the media? Is the digestive tract still able to 'defecate' ex vivo?

– A reference for the use of N-acetyl cysteine and sodium citrate as anti-phototoxicity agents would be helpful.

– Video 20 caption states that egg chamber rotation starts at Stage 5. My understanding is that rotation starts at Stage 1 (Cetara, 2014).

*Reviewer #2 (Recommendations for the authors):*

1) Authors presented the different steps of the dissection procedure in Figure 1 and Video 1 where they removed the abdomen, the Malpighian tubules, the ovaries, and the crop. What is the argument for not keeping them attached or near the intestine?

Keeping these tissues would limit the handling of the gut and induce less stress to the tissue. In addition, communication between the gut and the ovaries could be preserved. This is further relevant as the coculture with ovaries presented in Figure 2A shows a decrease cell death in explanted intestines. Authors could explain these additional steps.

2) Authors explained the negative effects of peristaltic movements on the intestine upon live-imaging to justify the use of the peristalsis inhibitor isradipine.

Authors could comment the impact of peristalsis inhibition on the normal gut functions.

3) To test the GFP expression in explanted intestines in progenitors (Figure 2D-E), authors used 5 to 15 days old flies. Did you use the same protocol for in vivo experiments? Did you see any impact of the age on the GFP expression in explanted intestines?

The n could be increased to further validate the use of this system.

4) Figure 3: Nuclei seem bigger upon SDS treatment (Figure 3B) compared to control (Figure 3A) at the T0 and also after. This difference is not seen in the graphs presented in Figure 3D-E. By looking the figure 3B, it is difficult to see the increase the size of the nuclei of cell losing the GFP signal.

Figure 3D-E present only few progenitor cells examples. What is the proportion of cells differentiating upon SDS treatment? By looking at figure 3B 48h, it seems that all of them are differentiating.

Is cell differentiation could be directly assessed with specific markers?

Did the authors check the cell death in explanted intestines after SDS treatment? and compared with in vivo conditions? Is it possible that SDS treatment can have a negative impact on extended culture of intestines?

Did the authors check the cell death in other genetic conditions used in the paper? In figure 3-supplement 2B? and others?

These experiments could be a good control to make sure that the different treatments applied have no effect on the extended culture of the intestines.

*Reviewer #3 (Recommendations for the authors):*

1. In the paper, the authors have used esgTs to drive expression of RasG12V or stg and CycE and examined the ISC proliferation and analyzed asymmetric verse symmetric divisions. However, esg-Gal4 is expressed in both ISCs and EBs. Overexpression of stg and CycE in EBs can drive EBs to divide, so using the division of progeny to define the symmetric or asymmetric divisions is not very precise here and the symmetric division rate is exaggerated (Figure 4). In addition, it might be the reason that the authors observed the similar speed of the migration in the progeny of both symmetric and asymmetric divisions (Figure 7—figure supplement 1). Therefore, the ISC specific Gal4 should be used to drive expression of stg and CycE in ISCs and the division of their progenies will be examined.

2. Loss of N induces accumulation of ISCs or ee cells, but these phenotypes are not observed in these explanted midguts (Figure 3 —figure supplement 2). Is it possible that this protocol for culturing explanted midguts affects stem cell's identity?

---

## [Author Response]

Essential revisions:1) Either substantiate or revise interpretations of symmetric/asymmetric fate outcomes.Because the movies use guts without markers for terminal fates, distinguishing asymmetric and symmetric outcomes is particularly challenging. Reviewers 1 and 3 felt that the criteria used to classify fate outcomes as symmetric or asymmetric were not sufficient to justify exclusion of the other outcome. Can the authors justify their interpretations, for instance by using midguts that express enteroblast- and enteroendocrine-specific markers? Alternately, if supporting evidence would not be available in a reasonable time frame, the authors are instead asked to describe the data in a manner that does not assign fate outcomes to division events that are ambiguous.

This is a reasonable criticism and we’ve done our best to address it. The suggestion to use enteroblast and enteroendocrine reporters (cell fate markers) along with the *esg^TS^* driver and *UAS-Ras^G12V^* or *UAS-stg UAS-CycE*, is not feasible within a reasonable time, as it would require us to develop and combine several new reporter lines.

As an alternative, we therefore tested an ISC-specific driver (*esg-Gal4 UAS-2XEYFP; Su(H)GBE-Gal80 tub-Gal80^TS^*) combined with UAS-YFP and the *His2Av.mRFP* nuclear marker to allow us to directly identify daughter cells as ISCs or EBs. Unfortunately, the combination of this ISC-specific driver and the *His2Av.mRFP* nuclear marker resulted in flies whose intestines were extremely prone to breaking during explant culture. These flies were also susceptible to infection, often leading to ex vivo culture contamination. These problems were observed in 6 separate live-imaging sessions with flies from 3 separate crosses. We did not observe these issues with any other genotype we tested, including the original ISC-specific driver and the *His2Av.mRFP* parent lines. Thus, unfortunately, we were not able to gather additional data to substantiate our assignments of symmetric vs. asymmetric fate outcomes. Given this, we have taken the alternative approach and modified our description of the lineages using the more general, conservative terms “co-dividing” and “non-co-dividing” instead of “symmetric” and “asymmetric” respectively (lines 277-283 and 301-307). Essentially, we have now classified cell divisions in the lineages based on their observed behavior, rather than their presumed fate outcomes. We do mention our expectation that “co-dividing” cells represent ISC duplications, but this is clearly presented as a likely possibility, not a definitive result.

2) Clarify the use of nuclear size as a measure of enteroblast differentiation.Reviewers 1 and 2 had questions about how nuclear size was used as a measure of enteroblast differentiation. For Figure 3 and Figure 4 movies, can the authors split the esg>GFP and his::RFP channels so that the nuclei are visible? This is critical to see nuclear size. For Figures 5-7 movies, only esg>GFP(nls?) is shown; can the his::RFP channel also be included? This is critical to evaluate that loss of GFP signal is not due to the trivial reason that the cell moved out of view. Can the authors provide data about the frequency of cell differentiation in the SDS movies? Can they provide additional examples of nuclear measurements for Figure 6E-H and Figure 7D-E (only one example of each type is shown)?

To better show nuclear size of progenitor cells in control and SDS-treated intestines, we have added a new video (Video 7, described at lines 183-185, 206-208, and 907-909) showing a close up of the cells described in Figure 3D-E. In this video, both individual channels (*nlsGFP* and *His2Av.mRFP*) and their merge are shown.

Regarding Video 18 (previously Video 16), to show that the non-dividing daughter of the first division event is differentiating (losing nlsGFP expression and growing in size), please refer to Author response image 1, which shows in more detail the cells of the lineage at 46h after imaging start. The *nlsGFP* and *His2Av.mRFP* of the 3 final cells in the lineage are here more clearly visible. The differentiating cell (#3) is still in the field of view as can be seen by the *His2Av.mRFP* signal. Please note its large nuclear size and low level of GFP expression.

Lastly, to quantify the frequency of cell differentiation in SDS movies, we have added a quantification of nuclear growth and GFP intensity of all the clearly trackable *esg^+^* cells in the control and SDS-treated intestines that we imaged (Figure 3F-G). This data shows that many, but not all, *esg^+^* cells growing in size and begin to lose GFP expression in response to SDS damage.

**Author response image 1. sa2fig1:** 

3) Provide additional information about organ viability in general and about ex vivo vs in vivo damage response.Reviewers 1 and 2 raised some questions about organ viability. Can the authors provide data for the typical viability of midguts subjected to confocal imaging at frame rates more frequent than 1 hour? Can they comment on how trachea and muscle may be altered by ex vivo culture conditions? The authors develop an ingenious SDS pulse-chase injury protocol; can the authors compare the ex vivo damage response shown in Figure 4 to an in vivo response to the same pulse-chase treatment?

If imaging parameters (*e.g.* laser power, dwell-time, etc.) are optimized, it is possible to image intestines for 48h at higher frame-rates (*e.g.* 15 min). We have added a new movie (Video 23, described at lines 951-953) captured using a Leica SP8 confocal microscope to show this possibility. There are, however, some limitations: (1) acquisition with scanning confocal microscopes is slow and when imaging several intestines at the same time this can limit the framerate; (2) the file size of 48h movies captured at high frame rates is substantial and requires a powerful data storage infrastructure for storage and analysis. These considerations have been added to the Results section, lines 139-145.

Trachea are severed when the intestine is detached from the abdomen and ovaries (line 96). The cell death marker, NucGreen, can be seen incorporated in tracheal nuclei (line 112). The visceral muscle, on the other hand, is healthy after explant and resumes contracting with regular peristalsis after the isradipine wears off. This is common in 48-72 hour cultures. A new video has been included to show regular peristaltic movements after 3 days ex vivo (Video 3, described at lines 894-896).

We have added a new supplementary figure showing the effect of our SDS protocol in vivo (Figure 3 —figure supplement 1). The data supports our ex vivo observations. It must be noted, however, that the in vivo and ex vivo conditions are not perfectly matched: the explanted intestines still contain low amounts of SDS, which continues to damage the epithelium while imaging, while in vivo flies were fed clean food after the SDS treatment. Nonetheless, after SDS feeding, intestines in vivo also show a large number of differentiating cells and late onset cell proliferation. These results are now discussed in greater detail in the main text, lines 210-220 and 842-852.

Reviewer #1 (Recommendations for the authors):– The terms motility and cell contact are used to discuss the data in Figure 7. However, since non-esg+ cells in the tissue are not visible, it is unclear whether the analyzed cells are autonomously motile or whether they are passively moving with or displaced by other cells. Similarly, since only cell nuclei are visible, cell-cell contact cannot be ascertained (e.g. lines 417-418). Can the authors replace "motility" and "cell contact" with terms that encompass these ambiguities, or else provide information that substantiates these interpretations?

Most non-*esg^+^* cells in the tissue are enterocytes. We did not observe enterocytes moving in the tissue, except for minor rearrangements (< 5µm) due to the loss of a neighboring cell. Indeed, enterocyte nuclei are a good reference for image registration algorithms. Contrary to this, *esg^+^* cells are highly motile in the presence of tissue damage (see also observed by Hu *et al.*, 2021) or when stimulated by *Ras^G12V^* or *stg* and *CycE*.

On the point of cell motility, since *esg^+^* cells have very high nucleus/cytoplasm ratio, using nuclear distance is a good approximation for measuring cell motility.

– The text states that the dissection technique used by the authors (which is nicely illustrated in Video 1) is an improvement over the standard technique. Can the authors revise the text to clarify this comparison, for instance by describing what the standard technique is?

We do not make any mention of a “standard” technique. It can be expected that any one individual researcher may dissect intestines in their own unique way. As dissection is a crucial step during which it’s easy to accidentally damage the intestine, we devised an optimized dissection protocol to reduce variation between users and improve midgut survival and integrity in culture.

– In Figures4-7, I'm confused why GFP fluorescence appears to localize to cell nuclei, not the cytoplasm, given that the genotype in the figure caption does not indicate a GFP:nls. Why is GFP used to measure nuclear size rather than the (presumably more accurate) His::RFP? Why is the RFP channel not shown?

The samples used for figures 3-7 do express a nuclear GFP (*nlsGFP*). As such, the signal overlaps with *His2Av.mRFP* and nuclear sizes measured with the two markers are equivalent. The Figure captions have been corrected accordingly for clarity. The RFP channel was not displayed in some panels to more easily show progenitor cells, which would otherwise be harder to visualize intermixed with the larger RFP+ enterocytes. The *His2Av.mRFP* signal is usually much brighter in the large enterocytes than in the smaller progenitor cells, compounding this problem. Moreover, due to the major tissue rearrangements induced by *Ras^G12V^* or *stg* and *CycE*, *esg^+^* cells can frequently overlap with enterocytes, in which case the RFP-labeled enterocyte and progenitor cell nuclei become indistinguishable.

– Which asymmetric divisions were analyzed in Figure 7 (all 25 asymmetric divisions, or a subset)? Can the authors revise Figure 7D to show individual data points rather than averages?

All asymmetric divisions were included in the analysis. We also tested the differences in internuclear distance using only those asymmetric divisions for which we observed the non-diving cell losing GFP expression. The same significant difference to symmetric divisions was observed, albeit with higher p values due to the reduced number of samples, but still below the 0.05 significance threshold.

Figure 7D has been modified as requested.

– Peristaltic contractions: Do guts remain viable if isradipine is not included in the media? Is the digestive tract still able to 'defecate' ex vivo?

Peristaltic movements result in more frequent midgut breaks, usually in the posterior section. However, undamaged areas (*i.e.* distant several hundred microns from the break site) remain viable, with regular muscle movements and normally behaving epithelial cells.

It must be noted that isradipine has no negative impact on muscle viability. After ~3 days of culture, the effect of isradipine usually wears off. When this occurs, the visceral muscle resumes regular peristaltic movements (Video 3 and lines 130-132). In the absence of isradipine, or when its effect wears off, midguts can indeed be seen “defecating”. However, this is undesirable as it results in the culture medium being contaminated by enteric bacteria, which quickly grow and result in the death of the cultured intestines.

– A reference for the use of N-acetyl cysteine and sodium citrate as anti-phototoxicity agents would be helpful.

One of the main causes of phototoxicity is the generation of reactive oxygen species (see Icha et al., 2017). Adding antioxidants to the culture medium is therefore a viable strategy to reduce the risk of phototoxicity. N-acetyl cysteine is an antioxidant widely used in cell culture (see Ezerina et al., 2018). Citrate is also known to have antioxidant properties (see Wu et al. 2019) and to be present in *Drosophila* hemolymph at detectable levels (see Echalier, “*Drosophila* Cells in Culture”). We have added these citations to the text in the appropriate places in the Results section (lines 132-135).

– Video 20 caption states that egg chamber rotation starts at Stage 5. My understanding is that rotation starts at Stage 1 (Cetara, 2014).

We apologize for the mislabeled Video. We meant to say that a rotating stage 4 egg chamber was seen starting to elongate, with elongation being a sign of progression to stage 5. As for egg chamber rotation, we did observe this phenomenon also in egg chambers at earlier stages. We have corrected the text accordingly (lines 378-382, 832-835, and 947-950).

Reviewer #2 (Recommendations for the authors):1) Authors presented the different steps of the dissection procedure in Figure 1 and Video 1 where they removed the abdomen, the Malpighian tubules, the ovaries, and the crop. What is the argument for not keeping them attached or near the intestine?Keeping these tissues would limit the handling of the gut and induce less stress to the tissue. In addition, communication between the gut and the ovaries could be preserved. This is further relevant as the coculture with ovaries presented in Figure 2A shows a decrease cell death in explanted intestines. Authors could explain these additional steps.

The main reason these structures are removed is to better access the intestine for imaging. When midguts are separated from neighboring organs and the abdominal cuticle, handling is easier and any region of the intestine is readily accessible. Consequently, imaging is also significantly clearer as ovaries and other organs could easily overlap with the midguts. Malpighian tubules also frequently wrap around midguts and stick to forceps during transfer from dissection dishes to imaging chambers. The crop and hindgut are also removed as they are easily damaged, which results in the release of enteric bacteria. We have expanded the relevant Material and Methods section (lines 604-608) to better explain these reasons.

2) Authors explained the negative effects of peristaltic movements on the intestine upon live-imaging to justify the use of the peristalsis inhibitor isradipine.Authors could comment the impact of peristalsis inhibition on the normal gut functions.

We have added comments on the possible impact of peristalsis inhibition in the Discussion section (lines 480-486).

3) To test the GFP expression in explanted intestines in progenitors (Figure 2D-E), authors used 5 to 15 days old flies. Did you use the same protocol for in vivo experiments? Did you see any impact of the age on the GFP expression in explanted intestines?The n could be increased to further validate the use of this system.

All flies used for both in vivo or ex vivo experiments shown were aged 5 to 15 days post-eclosion. We did not observe any obvious differences in GFP expression in either younger (1-5 days post-eclosion) or older (20-30 days post-eclosion) flies. Older flies did, however, show tissue alterations such as dysplasia at the time of dissection, while younger ones had usually a smaller intestinal diameter and strong luminal autofluorescence.

4) Figure 3: Nuclei seem bigger upon SDS treatment (Figure 3B) compared to control (Figure 3A) at the T0 and also after. This difference is not seen in the graphs presented in Figure 3D-E. By looking the figure 3B, it is difficult to see the increase the size of the nuclei of cell losing the GFP signal.

We have added quantifications of nuclear size showing that some *esg^+^* cells have a size larger than control cells already at T0. We have also added a new video (Video 7) that more clearly shows the nuclear size and GFP signal of cells from Figure 3D-E.

Figure 3D-E present only few progenitor cells examples. What is the proportion of cells differentiating upon SDS treatment? By looking at figure 3B 48h, it seems that all of them are differentiating.

We have now quantified the GFP levels of more cells at both 0h and 48h. This shows that many cells, but not all, lose GFP expression. This new data is shown in Figure 3G, and discussed on lines 206-208.

Is cell differentiation could be directly assessed with specific markers?

Cell differentiation could be assessed using specific fluorescent reporters. Unfortunately, available reporters fluoresce at wavelengths that overlap with either the GFP or RFP reporters used to mark cells in our driver line, and so we could not use these reporters in our experiments without losing the ability to track cells.

Did the authors check the cell death in explanted intestines after SDS treatment? and compared with in vivo conditions? Is it possible that SDS treatment can have a negative impact on extended culture of intestines?

Any kind of damage, be it chemical or mechanical, does result in reduced survival ex vivo. This also includes SDS feeding, especially since, at the time of dissection, the intestines still contain SDS in the lumen and we do not allow flies time to completely flush the SDS from it. Mechanical damage like improper dissection or the puncture of the epithelium described in Figure 3 —figure supplement 1 also reduces gut viability.

Did the authors check the cell death in other genetic conditions used in the paper? In figure 3-supplement 2B? and others?

Unless the intestine was damaged during dissection (e.g. Figure 3 —figure supplement 3B and Video 12), we did not observe cell death with any of the genotypes we used. However, we did observe that the strong cell proliferation induced by either *Ras^G12V^* or *stg* and *CycE* results in extrusion of enterocytes in regions were cell crowding is high, as it is known to occur also in vivo (see Patel *et al.*, 2019).

Reviewer #3 (Recommendations for the authors):1. In the paper, the authors have used esgTs to drive expression of RasG12V or stg and CycE and examined the ISC proliferation and analyzed asymmetric verse symmetric divisions. However, esg-Gal4 is expressed in both ISCs and EBs. Overexpression of stg and CycE in EBs can drive EBs to divide, so using the division of progeny to define the symmetric or asymmetric divisions is not very precise here and the symmetric division rate is exaggerated (Figure 4). In addition, it might be the reason that the authors observed the similar speed of the migration in the progeny of both symmetric and asymmetric divisions (Figure 7—figure supplement 1). Therefore, the ISC specific Gal4 should be used to drive expression of stg and CycE in ISCs and the division of their progenies will be examined.

Overexpression of *stg* and *CycE* can indeed force some EBs to divide (see Kohlmaier et al. Oncogene 2015). We agree with the reviewer that our definition of symmetric and asymmetric divisions was not precise since cell identity cannot be assessed with certainty in these experiments. To address this issue, we did a number of live imaging trials using a stem cell specific Gal4 driver (see answers to editor above for details). However, these trials failed because those midguts were to fragile to image successfully. Therefore, we have changed our discussion in the text by using the more precise terms “co-dividing” and “non-co-dividing” cells to describe the divisions we previously called “symmetric” and “asymmetric” (see lines 277-283 and 301-307). These more conservative terms are accurate operational definitions, and make convey the same information without presuming unknown cell fate outcomes.

2. Loss of N induces accumulation of ISCs or ee cells, but these phenotypes are not observed in these explanted midguts (Figure 3 —figure supplement 2). Is it possible that this protocol for culturing explanted midguts affects stem cell's identity?

In fact we did observe the accumulation of small-sized GFP^+^ cells in damaged *N^RNAi^*-expressing intestines (please see Figure 3 —figure supplement 2B). These cells were proliferative and remained small in size, similar to what has been described in the literature using fixed samples. However, it’s possible that the 48h time-frame may not have been sufficient to observe the formation of large ISC/EE tumor masses in our experiments. In our previous work (see Patel et al., Nat Cell Biol. 2015) we noted a great deal of variability in the N- phenotype, which was dependent upon not only timing, but also background genotype, microbiota, and the specific genetic elements used to suppress Notch signaling.